# Calibration Enhanced Decision Maker: Towards Trustworthy Sequential Decision-Making with Large Sequence Models

**Haoyuan Sun**                                                    *sun-hy23@mails.tsinghua.edu.cn*
*Tsinghua Shenzhen International Graduate School,*
*Tsinghua University*

**Bo Xia**[*]                                                     *xiab21@foxmail.com*
*College of Electronic Science and Technology,*
*National University of Defense Technology*

**Yifu Luo**
*Tsinghua Shenzhen International Graduate School,*
*Tsinghua University*

**Tiantian Zhang**[*]                                            *zhang.tt@sz.tsinghua.edu.cn*
*Tsinghua Shenzhen International Graduate School,*
*Tsinghua University*

**Xueqian Wang**                                                  *wang.xq@sz.tsinghua.edu.cn*
*Tsinghua Shenzhen International Graduate School,*
*Tsinghua University*

**Reviewed on OpenReview:** *https: // openreview. net/ forum? id= b6WcxPEb48*

## Abstract

Offline deep reinforcement learning (offline DRL) has attracted considerable attention across various domains due to its ability to learn effective policies without direct environmental interaction. Although highly effective, the trustworthiness of agent concerns the community. Offline DRL can be categorized into three principal paradigms: model-based algorithms, model-free algorithms, and trajectory optimization. While extant research predominantly concentrates on calibration enhancement of model-based and model-free algorithms, calibration of trajectory optimization remains a rather rare topic. In this paper, we introduce the concept of Expected Agent Calibration Error (EACE), a novel metric designed to assess agent calibration. Furthermore, we rigorously prove its theoretical relationship to the state-action marginal distribution distance. Subsequently, we introduce the Calibration Enhanced Decision Maker (CEDM), which employs a binning executor to process feature distribution histograms as input for the large sequence model, thereby minimizing the state-action marginal distribution distance and enhancing the agent's calibration. A series of in-depth case studies of CEDM are carried out, with application on Decision Transformer, Decision ConvFormer, and Decision Mamba. Empirical results substantiate the robustness of EACE and demonstrate the effectiveness of CEDM in enhancing agent calibration, thereby offering valuable insights for future research on trustworthy sequential decision-making.

## 1 Introduction

*"A good decision is based on knowledge and not on numbers."*

*—— Plato*

---

[*]Corresponding authors: Bo Xia, Tiantian Zhang

Deep reinforcement learning, employing the trial-and-error mechanisms to learn and optimize specific reward signals, has emerged as a powerful learning strategy executing autonomous data collection across various domains, such as robotic motion control (Singh et al., 2022; Tang et al., 2024), autonomous driving (Kiran et al., 2021; Zhao et al., 2024), large language model fine-tuning (Ouyang et al., 2022; Guo et al., 2025), and so on. Furthermore, it is the aspiration of many roboticists to program a robot with a task in the evening and return the following morning to discover it capable of effectively solving the task. Then, offline deep reinforcement learning (offline DRL) has garnered considerable interest in the community (Gürtler et al., 2023), primarily on account of its capacity to acquire effective strategies without the need for interaction with the environment. Such capability is particularly advantageous in scenarios where the cost or risk associated with real-time environmental engagement is substantial. According to survey compiled by Prudencio et al., it can be classified into three primary aspects: *(1) model-based algorithms* (Luo et al., 2024), *(2) model-free algorithms* (Swazinna et al., 2022), *(3) trajectory optimizations* (Janner et al., 2021; Hu et al., 2024).

Especially, for the third aspect, trajectory optimization approaches reformulate reinforcement learning problems as sequence modeling tasks, leading to improved performance and generalization. A large proportion of trajectory-optimization approaches adopt high-capacity large sequence models, which effectively address several long-standing challenges in conventional reinforcement learning methods, including sample efficiency, credit assignment, and partial observability. Within this paradigm, Decision Transformer (Chen et al., 2021) stands as a pioneering contribution. As shown in Figure 1(d), Decision Transformer processes a sequence that is composed of states, past actions, and desired returns within an autoregressive framework, and subsequently infers actions aimed at achieving the specified return, with the Transformer functioning as the policy model. Furthermore, following the advancements in large sequence models, adopting more powerful backbone architectures for sequential decision-making has also drawn increasing attention, as exemplified by Decision ConvFormer (Kim et al., 2024) and Decision Mamba (Ota, 2024).

Meanwhile, trustworthiness of the agent has also standed as a paramount concern within the community (Yu et al., 2025). It is widely acknowledged that miscalibrated trust can lead to misuse of agents (Wei et al., 2025). Therefore, to foster trustworthy and reliable agents, it is imperative to emphasize their calibration. In the aforementioned taxonomy of offline DRL, the first two categories already have corresponding works that consider calibration. Calibrated model-based DRL (Malik et al., 2019) demonstrates a method for endowing agents with a calibrated world model that accurately represents true uncertainty and enhances planning in high-stakes scenarios. Calibrated Q-learning (Nakamoto et al., 2024) learns conservative value functions (Kumar et al., 2020) that are calibrated with respect to behavior policy. However, current research on trajectory optimization has never incorporated calibration considerations, which poses a risk of misuse in future real-world applications. In this study, we primarily concentrate on offline DRL algorithms with large sequence models (such as Transformer (Vaswani, 2017), MetaFormer (Yu et al., 2022) and Mamba (Gu & Dao, 2024)). Therefore, we raise the following question:

> **Major Question**
>
> *Can we enhance calibration of offline DRL algorithms that utilize large sequence models to further bolster their trustworthiness in future real-world applications?*

To address this question, it is essential to first define a more accurate metric for agent calibration. Drawing inspiration from the definition of Expected Calibration Error (Guo et al., 2017) in prior model calibration literature, we introduce the *Expected Agent Calibration Error (EACE)* as a rigorous metric to quantitatively characterize the calibration properties of agents. Nonetheless, implementing measurement in practical assessments is challenging due to safety concerns. Therefore, we further theoretically prove the relationship between the EACE and the state-action marginal distribution distance: *differences in the EACE are bounded by the total variation distance between the state-action marginal distributions of two agents; and under mild assumptions, this can be extended to the Wasserstein-1 distance.* In other words, a smaller Wasserstein-1 distance between the state-action marginal distributions of two agents indicates a smaller difference in their EACE values, implying superior agent calibration.

Based on these foreshadowings, we introduce the *Calibration Enhanced Decision Maker (CEDM)*. Unlike traditional approaches that rely solely on the "number" of returns-to-go as input for the large sequence model, CEDM incorporates a binning executor to process feature distribution histograms, thereby providing

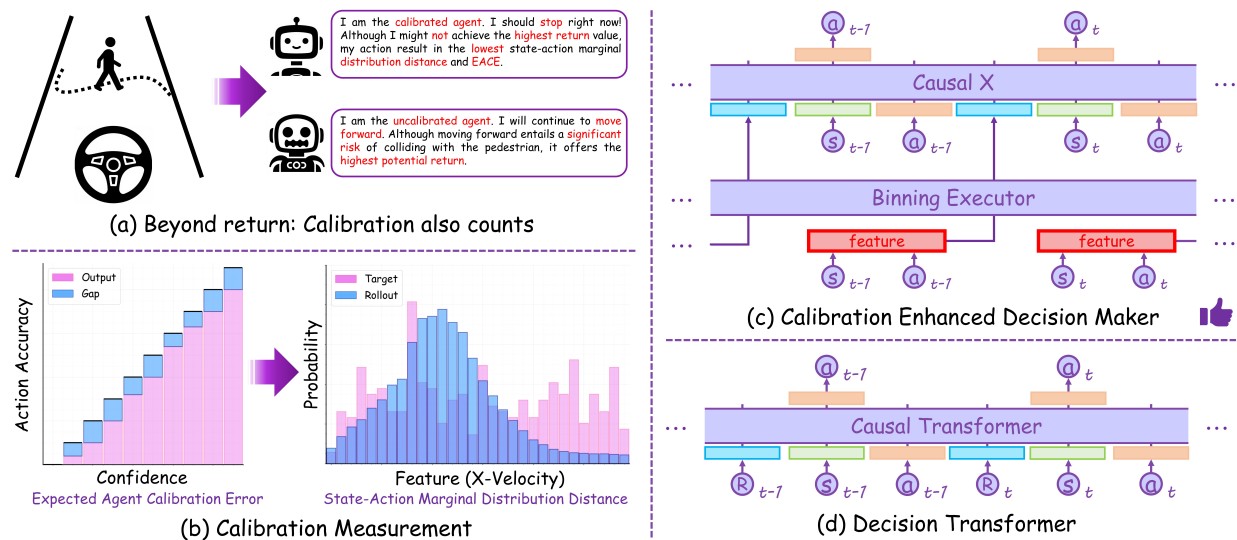

Figure 1: **(a) Motivation:** In real-world applications, especially in high-risk scenarios, trustworthiness of an agent's actions is also of great importance. **(b) Measurement:** We define the Expected Agent Calibration Error; furthermore, for practical applications, we establish its relationship with the state-action marginal distribution distance. **(c) Methodology:** We introduce the Calibration Enhanced Decision Maker paradigm, which employs a binning executor to process the distribution histograms as inputs to the large sequence model. **(d) Prototype:** Decision Transformer has been a groundbreaking approach in offline deep reinforcement learning that treats decision-making as a sequence modeling problem.

richer "knowledge" for the agent. Intuitively, CEDM enables the agent under training to more accurately model the state-action marginal distribution represented in the offline dataset. By minimizing the distance of the distributions, it consequently decreases the differences in the EACE, thereby enhancing agent calibration. Such paradigm is partially inspired by the previous studies on state-marginal matching (Lee et al., 2019; Ghasemipour et al., 2020): they conduct a qualitative assessment of distribution matching outcomes within designated state dimensions (e.g., $xy$-positions); furthermore, it can be extended to the velocity (e.g., $x$-velocity) to obtain richer "knowledge". Categorical Decision Transformer (Furuta et al., 2022) has shown the potential of a similar paradigm within the Transformer architecture and has explained its effectiveness from the perspective of hindsight information matching. Moreover, we demonstrate the applicability of such paradigm to a wider range of large sequence models, which reduces the distance between state-action marginal distributions and thereby enhances the agent calibration.

Furthermore, we perform case studies by applying the CEDM paradigm to *Decision Transformer* (Chen et al., 2021), *Decision ConvFormer* (Kim et al., 2024), and *Decision Mamba* (Ota, 2024). Based on a comprehensive review of these practical implementations, several conclusions can be drawn. Firstly, the CEDM paradigm effectively decreases the Wasserstein-1 distance between the state-action marginal distributions of the target rollout and the agent's behavior. Secondly, performance of the agent (e.g., final return) can be improved by utilizing the CEDM paradigm. Finally, application of the CEDM paradigm to ultra-long context sequence models (e.g., Mamba) may encounter limitations inherent in the offline dataset itself, providing insights for future efforts to deploy more advanced sequence models in decision-making tasks.

In summary, our contributions are as follows:

- We introduce the agent calibration metric: Expected Agent Calibration Error. Furthermore, to facilitate practical implementation, we theoretically establish the relationship between the EACE and the state-action marginal distribution distance.

- We introduce the Calibration Enhanced Decision Maker paradigm, which utilizes a binning executor to process distribution histograms as inputs to the large sequence model. This paradigm effectively reduces the state-action marginal distribution distance, thereby enhancing the agent's calibration. To the best of our knowledge, this is the first study on the enhancement of agent calibration in the field of offline DRL with large sequence models.

- We perform case studies, applying the CEDM paradigm to Decision Transformer, Decision ConvFormer, and Decision Mamba. Experimental results demonstrate the effectiveness of CEDM and provide insights into its future applications in more advanced sequence models.

## 2 Related Work

### 2.1 Offline DRL with Large Sequence Models

Decision Transformer (Chen et al., 2021) represents a pioneering stride in reinforcement learning, reframing the decision-making as sequence modeling using the Transformer architecture to predict optimal actions by processing trajectories of state, action, and returns-to-go. Such an innovative training paradigm has sparked a burgeoning trend in leveraging advanced large sequence models to address decision-making tasks. Decision ConvFormer (Kim et al., 2024) utilizes the architectural framework of MetaFormer (Yu et al., 2022), employing localized convolutional filtering as the token mixer, adeptly capturing the intrinsic local dependencies within the trajectories. Decision S4 (David et al., 2023) employs an RNN-like S4 (Gu et al., 2022) architecture for inference, while Decision Mamba (Ota, 2024) directly substitutes the attention mechanism with Mamba (Gu & Dao, 2024). Furthermore, this paradigm has also been increasingly applied in various domains, including robotic motion control (Gajewski et al., 2024), autonomous driving (Li et al., 2025), and large language model fine-tuning (Hu et al., 2023). Despite the highly promising results of these endeavors, none of them have addressed the issue of agent calibration, thereby posing a significant risk to their reliability and trustworthiness in further real-world applications.

### 2.2 Model Calibration

Model calibration refers to the alignment between a model's predicted probabilities and the true likelihood of the events (Guo et al., 2017). For example, if the model assigns a 70% confidence level to a prediction, we would expect the predicted outcome to occur in approximately 70% of instances in practical application. Unlike accuracy, calibration emphasizes trustworthiness of confidence, tackling challenges such as overconfidence. This guarantees probabilistic reliability, which is of paramount importance in high-cost and high-stakes domains. The concept of calibration has found purchase in a multitude of domains, such as autonomous driving (Fu et al., 2012), computer vision (Guo et al., 2017), natural language processing (Zhao et al., 2021), and so on.

In the domain of offline DRL, researchers have also made significant strides in enhancing calibration. In the survey (Prudencio et al., 2023), offline DRL can be classified into three primary aspects: model-based algorithms, model-free algorithms, and trajectory optimizations. Calibrated model-based reinforcement learning, as detailed in (Malik et al., 2019), provides agents with a refined world model that accurately reflects inherent uncertainty, thereby improving planning in high-stakes scenarios. Calibrated Q-learning (Cal-QL) (Nakamoto et al., 2024) leverages a modified CQL framework to ensure learned policy Q-values remain above a lower bound and below upper bounds for suboptimal reference policies, thereby calibrating the learned Q-values to a reasonable scale. They have shown promising applications of calibration in both model-based and model-free algorithms; however, there is still a lack of effort to enhance calibration in trajectory optimizations. In this study, we aim to provide insights into bridging this gap.

## 3 Preliminaries

### 3.1 Offline DRL with Trajectory Optimization

In the scenario of offline DRL, our target is to identify an optimal policy $\pi^*(a|s)$, maximizing the expected cumulative reward $\mathbb{E}[\Sigma_{t=0}^T r_t]$. The state $s_t$ and the action $a_t$ are dictated by the behavior policy; while the next state $s_{t+1}$ and the reward $r_t$ are determined by the transition dynamics function. This enables the construction of an offline trajectory dataset $\tau = \{(s_t, a_t, s_{t+1}, r_t)_i\}$, where $i$ presents the timestep in the episode. Trajectory optimization methods recast the goal as minimizing the action reconstruction loss due to the lack of interaction with the environment:

$$\mathcal{L}_\theta = \mathbb{E}_{(R,s,a)\sim\tau}[\frac{1}{T}\Sigma_{i=1}^T \mathcal{L}_{\mathrm{MSE/CE}}(\hat{a}_t; a_t)], \tag{1}$$

where $\hat{a}_t = \pi(\cdot|s_{t-K+1:t}, R_{t-K+1:t}, a_{t-K:t-1})$ donates the reconstruction action; $K$ is context length of the sequence model; $R_t = \Sigma_{t'=t}^T r_{t'}$ is the returns-to-go. During the testing phase, a target returns-to-go is manually predefined to encapsulate the intended performance benchmarks. Trajectories from last $K$ timesteps are input into the model, which subsequently generates an action for the current timestep. Based on this, the environment provides the next state and reward, after which the returns-to-go is updated accordingly. Then, these new elements are also input into the model.

### 3.2 Expected Calibration Error

Expected Calibration Error (Naeini et al., 2015) has been a common metric for describing the calibration performance of models, which quantifies the discrepancy between the model's predicted confidence and its actual accuracy. A model is deemed as *"perfectly calibrated"* when there is a precise alignment between its confidence levels and its empirical performance:

$$\forall p \in [0,1], \mathbb{P}(\hat{Y} = Y|\hat{P} = p) = p,$$

where $\hat{Y}$ is the output of network prediction and $\hat{P}$ is its associated confidence (i.e. probability of correctness). Furthermore, the Expected Calibration Error (ECE) is defined as discrepancy between the empirical accuracy and the corresponding confidence level:

$$\mathrm{ECE} = \mathbb{E}_{\hat{P}}\left[\left|\mathbb{P}\left(\hat{Y} = Y|\hat{P} = p\right) - p\right|\right]. \tag{2}$$

## 4 Methodology

In this section, we aim to address the challenge of enhancing agent calibration of offline DRL algorithms that utilize large sequence models, thereby improving their trustworthiness. Firstly, we introduce a metric called Expected Agent Calibration Error to describe the agent calibration. Furthermore, we theoretically prove that the difference between Expected Agent Calibration Error is bounded by the total variation distance and the Wasserstein-1 distance (under mild assumptions) between the learned policy and the hidden policy. Building upon them, we introduce Calibration Enhanced Decision Maker, which further leverages the distribution histograms as input for the large sequence model, ultimately yielding the calibration enhanced agent.

### 4.1 Expected Agent Calibration Error

To address the aforementioned challenge, the first step is to formulate a precise description of agent calibration. Inspired by the concept of "perfect calibration" and incorporating the inherent mechanisms of offline DRL, we formally define the concept of "perfect policy calibration" in Definition 1:

**Definition 1.** *Let $(\Pi, d)$ be a metric space, where $\Pi$ denotes the space of all stochastic policies, and $d : \Pi \times \Pi \to \mathbb{R}_{\geq 0}$ is a policy distance metric. Defining $U = \mathbb{I}\left[dis\left(\pi_1, \pi_2\right) \leq \delta\right]$, where $\mathbb{I}(\cdot)$ is the the indicator function, $U = 1$ denotes that the distance between policy $\pi_1$ and policy $\pi_2$ is smaller than the threshold $\delta$. Hence, for the agent (model), a perfect calibration (with full confidence) can be expressed as:*

$$\forall p \in [0,1], \mathbb{P}\left(U = 1|\hat{P} = p\right) = p. \tag{3}$$

Intuitively, "perfect policy calibration" refers to the scenario: two policies are sufficiently similar with the probability $p$; and exactly a proportion $p$ of practical predictions are indeed sufficiently similar.

Building upon this and drawing inspiration from ECE (Guo et al., 2017), we define the Expected Agent Calibration Error in Definition 2. Intuitively, it describes the average discrepancy between the agent's true probability $p$ of "U=1" and the stated probability $\hat{P}$ when it asserts that "I am $\hat{P}$ confident that U=1":

**Definition 2** (Expected Agent Calibration Error (EACE)). *Define the miscalibration of the agent by computing the expectation of calibration error over predicted confidence $\hat{P}$:*

$$\text{EACE} = \mathbb{E}_{\hat{P}} \left[ |\mathbb{P}(\text{U} = 1 | \hat{P} = p) - p| \right]. \tag{4}$$

Nevertheless, we find that although EACE is theoretically complete and grounded, it is computationally impractical in real-world applications. Notably, in high-risk scenarios, particularly when the agent's true probability $p$ is relatively low, such measurement become exceedingly costly, as the estimation demands a substantial number of rollouts under the policy.

Furthermore, as illustrated in Figure 1(d), in the context of offline DRL with large sequence models, the input tokens generally consist of states, actions, and their associated returns-to-go. Therefore, the subsequent action is essentially determined by the knowledge embedded in the preceding states and the preceding actions. This further inspires us to delve deeper into the relationship between the EACE and the state-action marginal distribution distance. Ultimately, we elaborate on this relationship in Theorem 1. Specifically, the differences in EACE are bounded by the total variation distance between the state-action marginal distributions of two agents, as formally expressed in Theorem 1.

**Theorem 1.** *Suppose $\pi_{\theta_1}(a|s)$ and $\pi_{\theta_2}(a|s)$ are separately two policies of two agents, $\rho^{\pi_{\theta_1}}(s, a)$ and $\rho^{\pi_{\theta_2}}(s, a)$ are the state-action marginal distributions of the two agents, then:*

$$\text{EACE}(\pi_{\theta_1}) - \text{EACE}(\pi_{\theta_2}) \leq \mathbb{E} \left[ 4 \cdot \text{TV}\big(\rho^{\pi_{\theta_1}}(s, a), \rho^{\pi_{\theta_2}}(s, a)\big) \right], \tag{5}$$

*where $\text{TV}(\cdot)$ is the total variation distance.*

*Proof.* Before presenting a detailed proof of Theorem 1, we first introduce two lemmas. The derivation of Lemma 1 primarily utilizes vector transformations and the absolute value inequality; and the derivation of Lemma 2 primarily relies on the Hölder's inequality. Their detailed proofs are provided in Appendix A.

**Lemma 1.** *Suppose $a$, $b$, $c$ are three vectors, then we have that:*

$$\langle |a - b|, 2b \rangle - \langle |a - c|, 2c \rangle \leq \langle b + c + |a - b| + |a - c|, |b - c| \rangle. \tag{6}$$

**Lemma 2.** *$a$ and $b$ are two vectors, and each of their terms is nonnegative. Then, we can get:*

$$\langle a, b \rangle \leq \langle \|a\|_1, \|b\|_\infty \rangle, \tag{7}$$

*where $\|a\|_1$ represents the $L_1$-norm of vector $a$ and $\|b\|_\infty$ represents the $L_\infty$-norm of vector $b$.*

We now proceed to the proof of Theorem 1.

For any state-action marginal distribution, we set that $\hat{P} = p = \rho^{\pi_\theta}(s, a)$ without loss of generality. As $\rho^{\pi_\theta}(s, a) = \rho^{\pi_\theta}(s) \cdot \pi_\theta(a|s)$; and given policy $\pi$, the action is sampled with the policy $\pi_\theta(a|s)$. Thus, we have:

$$\begin{aligned} \text{EACE}(\pi_\theta) &= \mathbb{E}_{\hat{P}} \left[ \left| \mathbb{P}(\text{U} = 1 | \hat{P} = p, A = a) - p \right| \right] \\ &= \mathbb{E}_{\rho^{\pi_\theta}(s,a)} \left[ \left| \mathbb{P}(\text{U} = 1 | \hat{P} = \rho^{\pi_\theta}(s, a), A = a) - \rho^{\pi_\theta}(s, a) \right| \right] \\ &= \mathbb{E} \left[ \sum \rho^{\pi_\theta}(s, a) \left| \mathbb{P}(\text{U} = 1 | \hat{P} = \rho^{\pi_\theta}(s, a), A = a) - \rho^{\pi_\theta}(s, a) \right| \right]. \end{aligned} \tag{8}$$

We then abbreviate the conditional distribution $\mathbb{P}(\text{U} = 1 | \hat{P} = \rho^{\pi_\theta}(s, a), A = a)$ as $\rho^\pi$. Then, we have:

$$\text{EACE}(\pi_\theta) = \mathbb{E} \left[ \langle |\rho^\pi - \rho^{\pi_\theta}(s, a)|, \rho^{\pi_\theta}(s, a) \rangle \right], \tag{9}$$

where $|\rho^\pi - \rho^{\pi_\theta}(s,a)|$ and $\rho^{\pi_\theta}(s,a)$ are vectors and $\langle |\rho^\pi - \rho^{\pi_\theta}(s,a)|, \rho^{\pi_\theta}(s,a)\rangle$ represents the inner product of $|\rho^\pi - \rho^{\pi_\theta}(s,a)|$ and $\rho^{\pi_\theta}(s,a)$. Further, let's compare the EACE of two agents $\theta_1, \theta_2 \in \Theta$:

$$\text{EACE}(\pi_{\theta_1}) - \text{EACE}(\pi_{\theta_2}) = \mathbb{E}\Big[\langle |\rho^\pi - \rho^{\pi_{\theta_1}}(s,a)|, \rho^{\pi_{\theta_1}}(s,a)\rangle - \langle |\rho^\pi - \rho^{\pi_{\theta_2}}(s,a)|, \rho^{\pi_{\theta_2}}(s,a)\rangle\Big]. \tag{10}$$

According to Lemma 1, we can get:

$$
\begin{aligned}
&\text{EACE}(\pi_{\theta_1}) - \text{EACE}(\pi_{\theta_2}) \\
&\leq \mathbb{E}\Big[\Big\langle \frac{\rho^{\pi_{\theta_1}}(s,a) + \rho^{\pi_{\theta_2}}(s,a)}{2} + \frac{|\rho^\pi - \rho^{\pi_{\theta_1}}(s,a)| + |\rho^\pi - \rho^{\pi_{\theta_2}}(s,a)|}{2}, |\rho^{\pi_{\theta_1}}(s,a) - \rho^{\pi_{\theta_2}}(s,a)|\Big\rangle\Big].
\end{aligned} \tag{11}
$$

According to the Lemma 2, we can have that:

$$
\begin{aligned}
&\text{EACE}(\pi_{\theta_1}) - \text{EACE}(\pi_{\theta_2}) \\
&\leq \mathbb{E}\Big[\Big\langle \frac{\rho^{\pi_{\theta_1}}(s,a) + \rho^{\pi_{\theta_2}}(s,a) + |\rho^\pi(s,a) - \rho^{\pi_{\theta_1}}(s,a)| + |\rho^\pi(s,a) - \rho^{\pi_{\theta_2}}(s,a)|}{2}, |\rho^{\pi_{\theta_1}}(s,a) - \rho^{\pi_{\theta_2}}(s,a)|\Big\rangle\Big] \\
&\leq \mathbb{E}\Big[\|\rho^{\pi_{\theta_1}}(s,a) - \rho^{\pi_{\theta_2}}(s,a)\|_1 \cdot \Big\|\frac{\rho^{\pi_{\theta_1}}(s,a) + \rho^{\pi_{\theta_2}}(s,a) + |\rho^\pi(s,a) - \rho^{\pi_{\theta_1}}(s,a)| + |\rho^\pi(s,a) - \rho^{\pi_{\theta_2}}(s,a)|}{2}\Big\|_\infty\Big].
\end{aligned} \tag{12}
$$

Setting that:

$$m(\pi_{\theta_1}, \pi_{\theta_2}, \pi) = \Big\|\frac{\rho^{\pi_{\theta_1}}(s,a) + \rho^{\pi_{\theta_2}}(s,a) + |\rho^\pi(s,a) - \rho^{\pi_{\theta_1}}(s,a)| + |\rho^\pi(s,a) - \rho^{\pi_{\theta_2}}(s,a)|}{2}\Big\|_\infty.$$

For the sake that each term of the distributions $\rho^{\pi_{\theta_1}}(s,a), \rho^{\pi_{\theta_2}}(s,a)$ and $\rho^\pi(s,a)$ are bounded in $[0,1]$; hence, it is evident that $m(\pi_{\theta_1}, \pi_{\theta_2}, \pi) \leq 2$. Therefore, we can get that:

$$
\begin{aligned}
\text{EACE}(\pi_{\theta_1}) - \text{EACE}(\pi_{\theta_2}) &\leq \mathbb{E}\left[2 \cdot \|\rho^{\pi_{\theta_1}}(s,a), \rho^{\pi_{\theta_2}}(s,a)\|_1\right] \\
&= \mathbb{E}\left[4 \cdot \text{TV}\left(\rho^{\pi_{\theta_1}}(s,a), \rho^{\pi_{\theta_2}}(s,a)\right)\right],
\end{aligned} \tag{13}
$$

which completes the proof. $\qquad\square$

Moreover, in practical applications, the Wasserstein-1 distance is often considered more applicable than the total variation distance, as it can better capture geometric properties of the distributions. Therefore, we extend the total variation distance in Theorem 1 to the Wasserstein-1 distance according to the inequality between them. Further defining the two policies as the anticipated ideal policy and the current policy, we can obtain the Proposition 1. It is also worth noting that our analysis is restricted to the discrete-only setting, under the structural assumption that the finite discrete support admits strictly positive inner point distance in the underlying metric space.

**Proposition 1.** *Suppose $\pi(a|s)$ is the anticipated ideal policy, $\pi_\theta(a|s)$ is the policy of the agent, $\rho^\pi(s,a)$ is the state-action marginal distribution of the anticipated ideal agent and $\rho^{\pi_\theta}(s,a)$ is the state-action marginal distribution of the current agent, then:*

$$\text{EACE}(\pi_\theta) - \text{EACE}(\pi) \leq \mathbb{E}\left[\frac{4}{d_{min}} \cdot \text{W}_1\big(\rho^{\pi_\theta}(s,a), \rho^\pi(s,a)\big)\right], \tag{14}$$

*where $\text{W}_1\big(\rho^{\pi_\theta}(s,a), \rho^\pi(s,a)\big)$ represents the Wasserstein-1 distance between $\rho^{\pi_\theta}(s,a)$ and $\rho^\pi(s,a)$; let $\rho^{\pi_\theta}(s,a) \in \mu$ and $\rho^\pi(s,a) \in \nu$, setting $\Omega = supp(\mu) \cup supp(\nu)$, $d_{min} = \inf_{\rho^{\pi_\theta} \neq \rho^\pi \in \Omega} \|\rho^{\pi_\theta} - \rho^\pi\|$.*

*Proof.* According to the work (Panaretos & Zemel, 2019), we introduce the following Lemma 3, which establishes relationship between the TV distance and the Wasserstein-1 distance.

**Lemma 3** ((Panaretos & Zemel, 2019)). *Setting $X, Y$ are finitely discrete random variables, and they are bounded; $X \in \mu$ and $Y \in \nu$, $\Omega = supp(\mu) \cup supp(\nu)$; $d_{min} = \inf_{x \neq y \in \Omega} \|x - y\|$, then we have that:*

$$\text{TV}(X,Y) \leq \frac{1}{d_{min}} \cdot \text{W}_1(X,Y). \tag{15}$$

Herein, let's consider the proof of Proposition 1:

According to Theorem 1, we have that:

$$\text{EACE}(\pi_\theta) - \text{EACE}(\pi) \leq \mathbb{E}\left[4 \cdot \text{TV}(\rho^{\pi_\theta}(s,a), \rho^\pi(s,a)\right].$$

According to Lemma 3, setting that $\rho^{\pi_\theta}(s,a) \in \mu$, $\rho^\pi(s,a) \in \nu$, $\Omega = supp(\mu) \cup supp(\nu)$, $d_{min} = \inf_{\rho^{\pi_\theta} \neq \rho^\pi \in \Omega} \|\rho^{\pi_\theta} - \rho^\pi\|$, we can derive that:

$$\begin{aligned}
\text{EACE}(\pi_\theta) - \text{EACE}(\pi) &\leq \mathbb{E}\left[4 \cdot \text{TV}(\rho^{\pi_\theta}(s,a), \rho^\pi(s,a)\right] \\
&\leq \mathbb{E}\left[\frac{4}{d_{min}} \cdot \text{W}_1\left(\rho^{\pi_\theta}(s,a), \rho^\pi(s,a)\right)\right],
\end{aligned} \tag{16}$$

which completes the proof. $\square$

In summary, under mild assumptions, the EACE discrepancy between the anticipated ideal policy and the current policy is bounded by the Wasserstein-1 distance between their state-action marginal distributions. Hence, we can consider minimizing the upper bound, specifically the Wasserstein-1 distance, to reduce their EACE discrepancy.

## 4.2 Calibration Enhanced Decision Maker

Based on the comprehensive theoretical analysis presented above, we introduce the Calibration Enhanced Decision Maker (CEDM) paradigm, which employs categorical state-action marginal distribution histograms as the input token to the large sequence model. As illustrated in Figure 1(c), a binning executor is utilized to transform the feature distribution into a categorical approximation of the original continuous distribution. Previous research has also explored the binning paradigm to enhance input information density: building upon prior work on the state-marginal matching (Lee et al., 2019; Ghasemipour et al., 2020), the Categorical Decision Transformer (Furuta et al., 2022) demonstrates the potential of binning distributions, particularly reward distributions, within the Transformer architecture. Moreover, as demonstrated by our theoretical analysis, the CEDM paradigm can effectively enhance the agent's calibration and is applicable to all large sequence models instead of Transformer only.

Furthermore, as Plato stated, "a good decision is based on knowledge and not on numbers". The CEDM paradigm equips the agent with richer "knowledge", rather than relying solely on the "number" of returns-to-go. In practical applications, agents can be endowed with different dimensions of "knowledge" according to the specific requirements of each task. For example, in autonomous driving, the planar velocity ($x$-velocity) should be prioritized for consideration; in obstacle-crossing robots, the vertical velocity ($z$-velocity) is also crucial; in robotic arms, the joint angular velocity is of primary importance; and in more elaborate applications, such as missile tracking or space robots, the acceleration should also be considered. The introduced CEDM paradigm enables further selection of the appropriate state-action marginal distribution for binning, thereby allowing customization to meet specific practical requirements.

## 5 Case Studies

This section presents case studies applying the CEDM paradigm to Decision Transformer (Chen et al., 2021), Decision ConvFormer (Kim et al., 2024), and Decision Mamba (Ota, 2024), with reports on final returns and the Wasserstein-1 distances between state-action marginal distributions to validate its effectiveness and enhancement on agent calibration. These three architectures represent three distinct levels of dependence on historical information within the decision-making process. We focus on the following three questions:

1) Can the CEDM paradigm effectively reduce Wasserstein-1 distances, thereby enhance the agent's calibration?

2) Can the CEDM paradigm improve the agent's return?

3) Would the improvements of the CEDM paradigm be affected by different degrees of dependence on historical information during the decision-making process?

Table 1: W1_Dis and return comparison of Calibration Enhanced Decision Transformer (CEDT) and Decision Transformer (DT). **Bolded text** indicates that the results of CEDT are superior to those of DT.

| | CEDT | | | | | DT | | | | |
|---|---|---|---|---|---|---|---|---|---|---|
| | **W1_Dis ↓** | | | | | | | | | |
| | Med | Med-Exp | Med-Rep | Average | Total | Med | Med-Exp | Med-Rep | Average | Total |
| HC | **0.19±0.01** | **0.68±0.29** | **0.31±0.03** | **0.39** | | 0.20±0.02 | 1.06±0.34 | 0.32±0.11 | 0.53 | |
| Hp | **0.14±0.02** | **0.13±0.03** | **0.26±0.18** | **0.18** | **1.19** | 0.14±0.02 | 0.13±0.03 | 0.70±0.18 | 0.32 | 1.39 |
| Wk | **0.31±0.08** | **0.08±0.02** | 1.46±0.80 | 0.62 | | 0.43±0.11 | 0.08±0.04 | 1.12±0.62 | 0.54 | |
| | **Return ↑** | | | | | | | | | |
| HC | 42.77±0.26 | **86.64±2.30** | **39.59±0.25** | **56.33** | | 42.88±0.29 | 84.05±2.52 | 39.09±0.39 | 55.34 | |
| Hp | **59.85±0.52** | 78.88±16.25 | **62.11±17.96** | **70.96** | **200.27** | 57.80±2.41 | 90.16±11.58 | 20.34±5.51 | 56.10 | 182.11 |
| Wk | **73.77±4.28** | **107.70±0.70** | **37.48±17.83** | **72.98** | | 72.87±6.22 | 107.49±1.11 | 31.66±14.25 | 70.67 | |

Table 2: W1_Dis and return comparison of Calibration Enhanced Decision ConvFormer (CEDC) and Decision ConvFormer (DC). **Bolded text** indicates that the results of CEDC are superior to those of DC.

| | CEDC | | | | | DC | | | | |
|---|---|---|---|---|---|---|---|---|---|---|
| | **W1_Dis ↓** | | | | | | | | | |
| | Med | Med-Exp | Med-Rep | Average | Total | Med | Med-Exp | Med-Rep | Average | Total |
| HC | **0.20±0.05** | **0.46±0.25** | **0.09±0.04** | **0.25** | | 0.22±0.06 | 0.50±0.29 | 0.24±0.06 | 0.32 | |
| Hp | **0.13±0.02** | **0.12±0.04** | **0.28±0.19** | **0.18** | **0.69** | 0.15±0.03 | 0.15±0.07 | 0.93±0.33 | 0.41 | 1.16 |
| Wk | **0.27±0.10** | **0.07±0.01** | **0.45±0.32** | **0.26** | | 0.46±0.13 | 0.23±0.31 | 0.60±0.22 | 0.43 | |
| | **Return ↑** | | | | | | | | | |
| HC | **42.92±0.15** | **88.51±1.78** | **41.17±0.53** | **57.53** | | 42.81±0.22 | 88.16±2.53 | 39.69±0.21 | 56.89 | |
| Hp | **62.67±1.62** | 75.06±8.86 | **75.14±13.39** | **70.96** | **204.75** | 59.13±3.28 | 76.57±19.08 | 30.84±24.95 | 55.51 | 189.08 |
| Wk | **76.09±2.56** | **108.05±0.76** | 44.64±19.10 | 76.26 | | 74.87±2.83 | 104.36±7.38 | 50.81±16.34 | 76.68 | |

## 5.1 Experimental Setup

In our case studies, experiments are conducted with the OpenAI Gym environment (Brockman et al., 2016), MuJoCo tasks (Todorov et al., 2012), which serve as the most typical benchmark for reinforcement learning. Specifically, we utilize the task of HalfCheetah (**HC**), Hopper (**Hp**), Walker2d (**Wk**). In terms of datasets, we utilize medium (**Med**), medium-expert (**Med-Exp**), and medium-replay (**Med-Rep**) datasets that are from the D4RL benchmark (Fu et al., 2020).More details are provided in Appendix C. To ensure consistency, we utilize the distributions of $x$-velocity throughout our study. Furthermore, to facilitate a unified representation of returns across tasks, we perform normalization on the scores:

$$\text{Return} = 100 \times \frac{\text{Score} - \text{Random Score}}{\text{Expert Score} - \text{Random Score}}, \tag{17}$$

where the terms "Random Score" and "Expert Score" are adopted from the D4RL benchmark. When selecting the binning executor, we utilize the same binning method as that used in the Categorical Decision Transformer (Furuta et al., 2022). For each dataset in every task, we identify the best trajectory; and the results are reported as the "mean ± standard deviation" over five random seeds. Additionally, we calculate the **Average** performance for each task across the three corresponding datasets, as well as the **Total** results

Table 3: W1_Dis and return comparison of Calibration Enhanced Decision Mamba (CEDMamba) and Decision Mamba (DMamba). **Bolded text** indicates that results of CEDMamba are superior to those of DMamba.

| | CEDMamba | | | | | DMamba | | | | |
|---|---|---|---|---|---|---|---|---|---|---|
| | **W1_Dis ↓** | | | | | | | | | |
| | Med | Med-Exp | Med-Rep | Average | Total | Med | Med-Exp | Med-Rep | Average | Total |
| HC | 0.21±0.05 | 0.89±0.27 | **0.11±0.04** | 0.40 | | 0.18±0.04 | 0.66±0.13 | 0.31±0.11 | 0.38 | |
| Hp | 0.15±0.03 | **0.10±0.02** | **0.39±0.23** | **0.21** | **0.86** | 0.14±0.03 | 0.15±0.02 | 0.65±0.18 | 0.31 | 1.33 |
| Wk | **0.30±0.05** | **0.06±0.02** | 0.39±0.15 | **0.25** | | 0.50±0.18 | 0.88±0.25 | 0.55±0.27 | 0.64 | |
| | **Return ↑** | | | | | | | | | |
| HC | 42.87±0.23 | 85.59±2.05 | **40.86±0.21** | **56.44** | | 42.97±0.12 | 86.73±1.29 | 39.26±0.54 | 56.32 | |
| Hp | 59.73±2.28 | 90.28±9.08 | **78.19±2.12** | **76.07** | **212.94** | 69.39±7.25 | 97.47±10.47 | 41.03±19.03 | 69.30 | 196.72 |
| Wk | **76.69±2.40** | **108.05±1.00** | **56.56±5.12** | 80.43 | | 70.47±7.21 | 87.20±5.48 | 55.63±7.22 | 71.10 | |

Figure 2: Visualizations of the distribution histograms for the highest-return trajectory of CEDT (*left*), CEDC (*middle*) and CEDMamba (*right*), presented separately for the three tasks across the three datasets.

of the average values. The training hyperparameters and implementation details are consistent with those reported in the original paper.

## 5.2 Comparison Results

In Table 1, Table 2, and Table 3, we present the comparative results of Wasserstein-1 distance and return by applying the CEDM paradigm to Decision Transformer, Decision ConvFormer, and Decision Mamba, respectively. Moreover, the three architectures represent distinct levels of dependence on historical information within the decision-making process: Decision ConvFormer primarily utilizes local information, while Decision Mamba emphasizes global information.

Firstly, we analyze the impact of the CEDM paradigm on the agents' Wasserstein-1 distances to address the first question. From an overall perspective, specifically in comparison to the "Total" values, it is indicated that the Calibration Enhanced Decision Makers effectively reduce the Wasserstein-1 distance across all architectural configurations relative to the baseline methods. To be specific, Table 1 demonstrates that CEDT

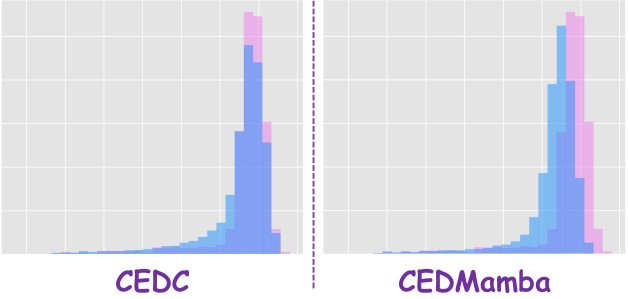

Figure 3: Example rollout (blue) visualizations of CEDC (*left*) and CEDMamba (*right*) on the HalfCheetah medium-expert dataset for the same target (violet).

achieves a reduction of 14.4% in the total Wasserstein-1 distance, decreasing it from 1.39 to 1.19; Table 2 indicates that CEDC attains a reduction of 40.5% in the total Wasserstein-1 distance, lowering it from 1.16 to 0.69; Table 3 reveals that CEDMamba accomplishes a reduction of 35.3% in the total Wasserstein-1 distance, reducing from 1.33 to 0.86. Consequently, the CEDM paradigm is empirically demonstrated to effectively reduce the Wasserstein-1 distance between state-action marginal distributions, thereby enhancing the agent calibration. Furthermore, to facilitate intuitive comprehension, we present representative visualizations of the distribution histograms for them, as depicted in Figure 2. Overall, the CEDM paradigm exhibits a good alignment with the distributions from the original dataset, thereby highlighting efficacy of CEDM paradigm.

Secondly, let's proceed to answer the second question. Although primary motivation of the CEDM paradigm is to enhance agent calibration, it is also crucial to evaluate its potential impact on the final return. By comparing the "Total" return values, it is evident that the Calibration Enhanced Decision Makers consistently outperform the baseline methods, yielding superior final returns across all architectural configurations. To be specific, Table 1 demonstrates that CEDT achieves an improvement of 10.0% in the total final return, elevating it from 182.11 to 200.27; Table 2 indicates that CEDC attains an improvement of 8.3% in the total final return, enhancing it from 189.08 to 204.75; Table 3 reveals that CEDMamba accomplishes an improvement of 8.2% in the total final return, raising it from 196.72 to 212.94. Notably, we observe that the improvements achieved by the CEDM paradigm are particularly pronounced on the Hopper-Medium-Replay dataset: specifically, CEDT exhibits an increase from 20.34 to 62.11; CEDC improves from 30.84 to 75.14; and CEDMamba demonstrates a rise from 41.03 to 78.19. According to the study (Ajay et al., 2023), the action trajectories in the Hopper-Medium-Replay dataset exhibit a propensity for diminished smoothness. Therefore, the CEDM paradigm is still capable of maintaining remarkable efficacy even when implemented on datasets of relatively lower quality.

Finally, we advance to answer the third question. Within the results, it is observed that the CEDM paradigm occasionally exhibits instances of failure, particularly in relation to CEDMamba. We argue that this phenomenon is attributable to the inherent ultra-long context capabilities of Mamba. In Figure 3, we present an example that demonstrates the differences in rollouts between CEDC and CEDMamba for the same target. For the CEDC, its convolutional architecture endows it with an enhanced capacity to capture local information, thereby exhibiting superior fitting performance within high-probability regions. However, CEDMamba predominantly emphasizes global information, thereby making its modeling of high-probability regions particularly vulnerable to biases stemming from the dataset. This further highlights two critical considerations for future real-world applications: firstly, the selection of an appropriate backbone should be tailored to the specific dataset and task; secondly, the quality of offline datasets plays a crucial role in shaping effectiveness of the CEDM paradigm.

# 6    Conclusion, Limitation, and Future Work

In this study, we focus on the crucial challenge of enhancing calibration in offline DRL algorithms that leverage large sequence models, with the aim of enhancing their trustworthiness in future real-world applications.

We begin by introducing the Expected Agent Calibration Error, a metric that describes the agent calibration. Furthermore, to facilitate practical assessments, we rigorously establish its theoretical relationship with the state-action marginal distribution distance: the EACE discrepancy between the anticipated ideal policy and the current policy is bounded by the Wasserstein-1 distance between the agent's state-action marginal distributions under the two policies. Subsequently, we introduce the Calibration Enhanced Decision Maker paradigm, which leverages a binning executor to process distribution histograms as inputs to the large sequence model. This paradigm aims at reducing the discrepancy between state-action marginal distributions, thereby enhancing agent's calibration. Moreover, we perform case studies, applying the CEDM paradigm to Decision Transformer, Decision ConvFormer, and Decision Mamba. Experimental results highlight effectiveness of the CEDM paradigm and shed light on its future application in more advanced large sequence models.

**Limitation.** Although the CEDM paradigm has demonstrated significant advancements in trustworthiness and has exhibited promising performance across simulation environments, its effectiveness in real-world applications necessitates further investigation and exploration.

**Future Work.** Going forward, efforts in the future could primarily focus on the following three points. Firstly, the applications of the CEDM paradigm to real-world scenarios warrant further exploration, particularly in the domains of robotic motion control and autonomous driving. Secondly, it is also worthwhile to further investigate the applications of diverse physical quantities beyond velocity, such as joint angular velocity and acceleration, within the CEDM paradigm. Lastly, future research should continue to investigate approaches in the domain of trajectory optimization that could enhance agent calibration.

### Broader Impact Statement

As stated in Section 4.2, the proposed CEDM paradigm allows flexible customization to meet specific practical requirements. Therefore, it holds great potential for real-world applications and is expected to be further extended to areas including robotics, autonomous driving, and aerospace engineering.

### Acknowledgments

This work was supported by the Natural Science Foundation of Shenzhen (No. JCYJ20230807111604008, No. JCYJ20240813112007010), the Natural Science Foundation of Guangdong Province (No. 2024A1515010003) and Cross-disciplinary Fund for Research and Innovation (No. JC2024002) of Tsinghua SIGS.

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

## A  Detailed Proof of Lemma 1 and Lemma 2

**Lemma 1.** *Suppose a, b, c are three vectors, then we have that:*

$$\langle |a-b|, 2b \rangle - \langle |a-c|, 2c \rangle \leq \langle b+c+|a-b|+|a-c|, |b-c| \rangle. \tag{18}$$

*Proof.* Consider the following inequation:

$$\langle |a-c|, b-c-|b-c| \rangle \leq \langle |a-b|, c-b+|c-b| \rangle \tag{19}$$

According to the nature of the absolute value, it is obvious that $b-c-|b-c| \leq 0$ and that $c-b+|c-b| \geq 0$, which shows the constancy of Equation (19) is obvious.
Transform Equation (19), we can get that:

$$\langle |a-c|, b \rangle + \langle |a-b|, b-|b-c| \rangle \leq \langle |a-c|, c+|b-c| \rangle + \langle c, |a-b| \rangle \tag{20}$$

Based on the absolute value inequality, we can get:

$$|a-b|-|b-c| \leq |a-c|; |a-c|+|b-c| \geq |a-b| \tag{21}$$

Hence, we have that:

$$\begin{aligned}
\langle |a-b|-|b-c|, b \rangle + \langle |a-b|, b-|b-c| \rangle &\leq \langle |a-c|, b \rangle + \langle |a-b|, b-|b-c| \rangle \\
&\leq \langle |a-c|, c+|b-c| \rangle + \langle c, |a-b| \rangle \\
&\leq \langle |a-c|, c+|b-c| \rangle + \langle c, |a-c|+|b-c| \rangle
\end{aligned} \tag{22}$$

Combine some items of the same kind:

$$\langle |a-b|-|b-c|, b \rangle + \langle |a-b|, b-|b-c| \rangle \leq \langle |a-c|+|b-c|, c \rangle + \langle |a-c|, c+|b-c| \rangle \tag{23}$$

Thus, we have that:

$$2\langle |a-b|, b \rangle - 2\langle |a-c|, c \rangle \leq \langle b, |b-c| \rangle + \langle c, |b-c| \rangle + \langle |a-b|, |b-c| \rangle + \langle |a-c|, |b-c| \rangle \tag{24}$$

Then, we have that:

$$\langle |a-b|, 2b \rangle - \langle |a-c|, 2c \rangle \leq \langle b+c+|a-b|+|a-c|, |b-c| \rangle, \tag{25}$$

which completes the proof. $\qquad\square$

For the proof of Lemma 2, we rely on the following fact, which is also denoted as the Hölder's inequality.

**Fact 1** (Hölder's inequality). *Set $p > 1, 1/p + 1/q = 1$, if $a_1, a_2...a_n$ and $b_1, b_2...b_n$ is nonnegative, then we have:*

$$\sum_{i=1}^{n} a_i b_i \leq \left( \sum_{i=1} a_i^p \right)^{\frac{1}{p}} \left( \sum_{i=1} b_i^q \right)^{\frac{1}{q}} \tag{26}$$

**Lemma 2.** *a and b are two vectors, and each of their terms is nonnegative. Then, we can get:*

$$\langle a, b \rangle \leq \langle \|a\|_1, \|b\|_\infty \rangle, \tag{27}$$

*where $\|a\|_1$ represents the $L_1$-norm of vector a and $\|b\|_\infty$ represents the $L_\infty$-norm of vector b.*

*Proof.* Setting p as $\infty$ and q as 1, then according to the Equation (26) (Hölder's inequality), we can have:

$$\sum_{i=1}^{n} a_i b_i \leq \|b\|_\infty \cdot \|a\|_1 \tag{28}$$

Thus, we have that:

$$\langle a, b \rangle \leq \langle \|a\|_1, \|b\|_\infty \rangle, \tag{29}$$

which completes the proof. $\qquad\square$

# B   Additional Visualizations of the Distribution Histograms

In this section, we provide more visualizations of the distribution histograms of four additional trajectories. In each figure, each row represents results on the specific dataset: medium as **Med**; medium-expert as **Med-Exp**; medium-replay as **Med-Rep**.

**CEDT:** Figure 4 (HalfCheetah), Figure 5 (Hopper), Figure 6 (Walker2d).

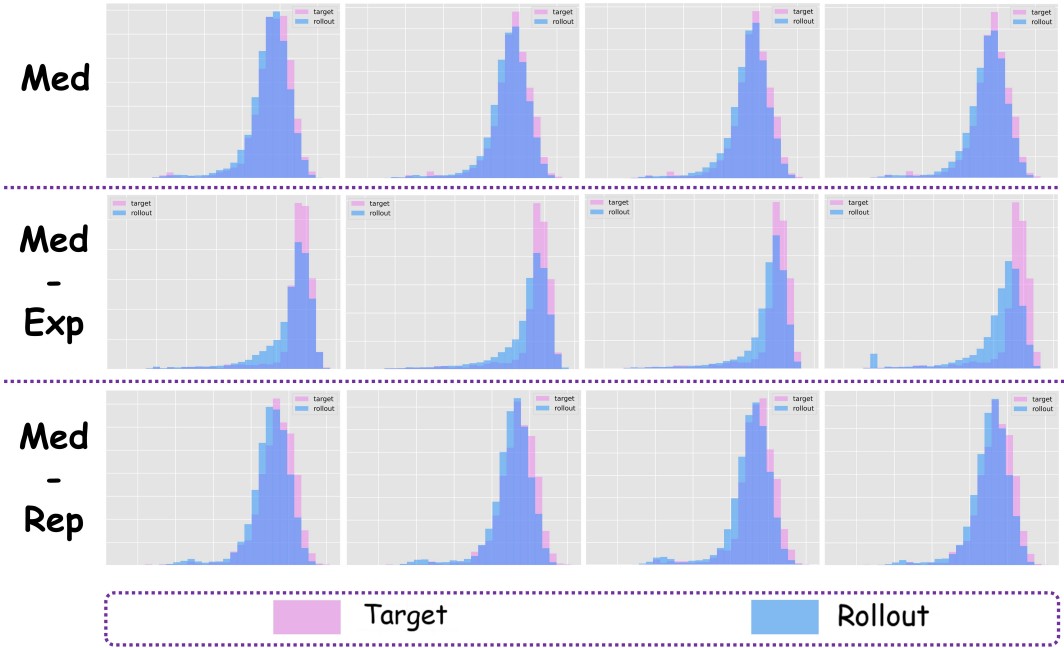

Figure 4: Within the **CEDT**, for the **HalfCheetah** task, additional visualizations of distribution histograms across three datasets.

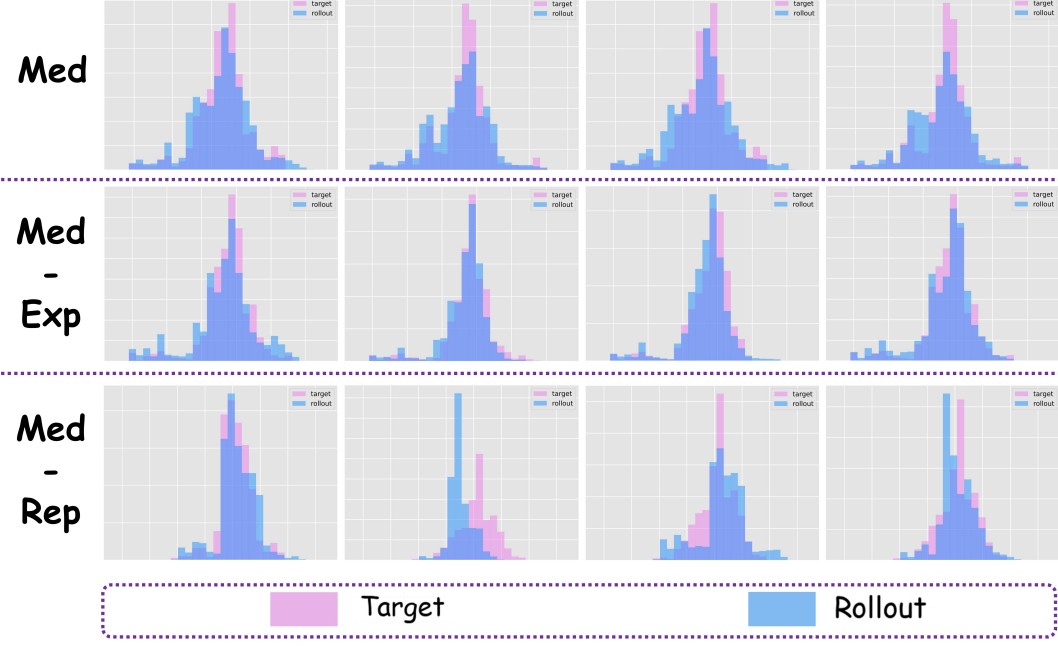

Figure 5: Within the **CEDT**, for the **Hopper** task, additional visualizations of distribution histograms across three datasets.

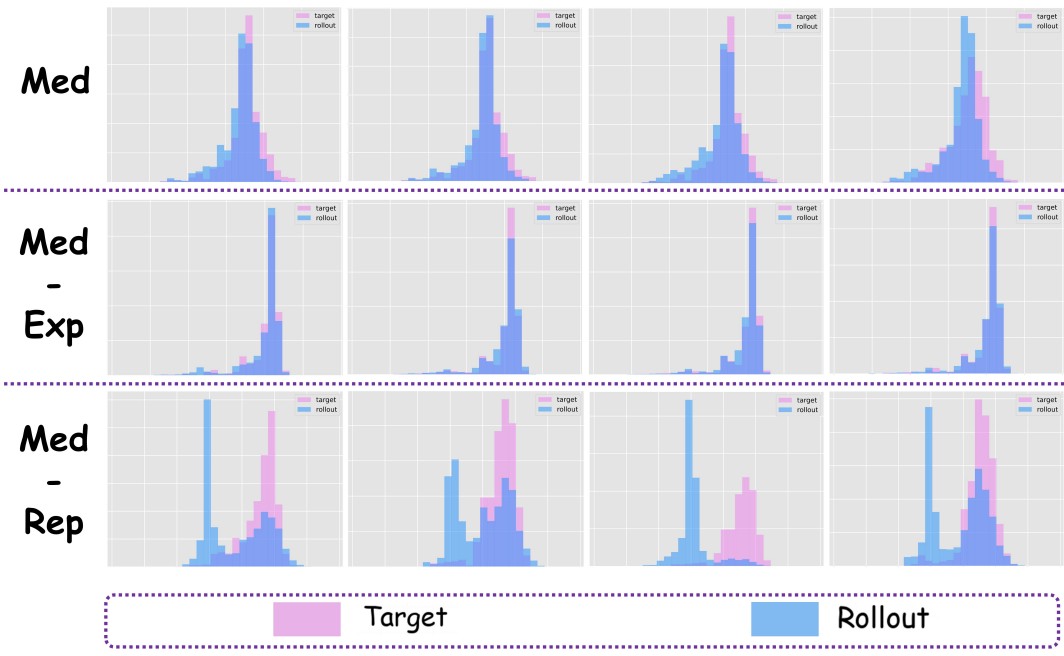

Figure 6: Within the **CEDT**, for the **Walker2d** task, additional visualizations of distribution histograms across three datasets.

**CEDC:** Figure 7 (HalfCheetah), Figure 8 (Hopper), Figure 9 (Walker2d).

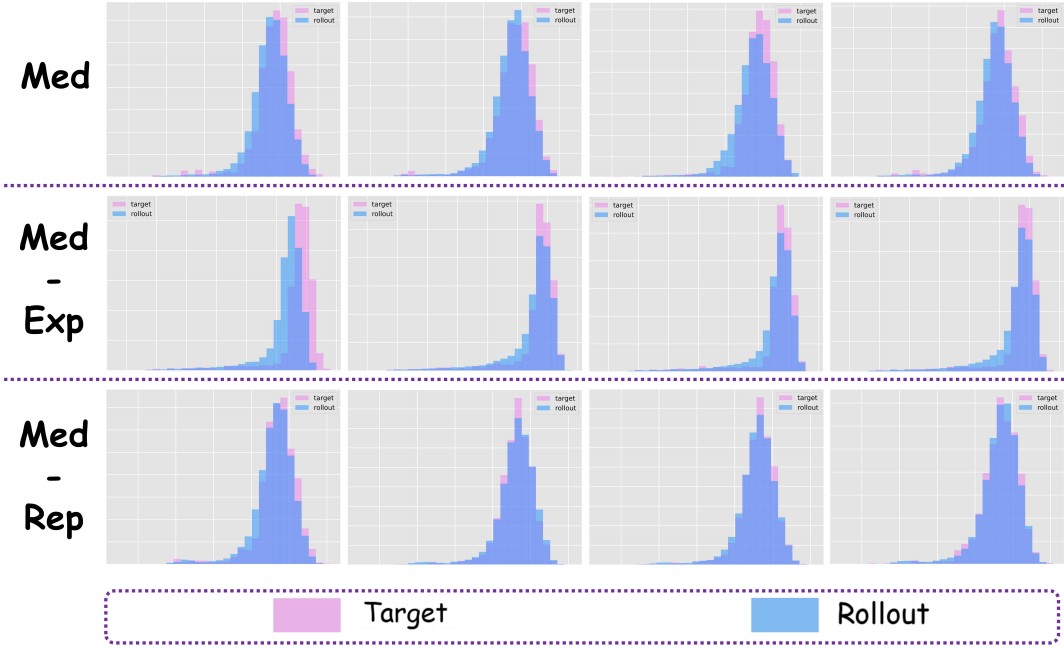

Figure 7: Within the **CEDC**, for the **HalfCheetah** task, additional visualizations of distribution histograms across three datasets.

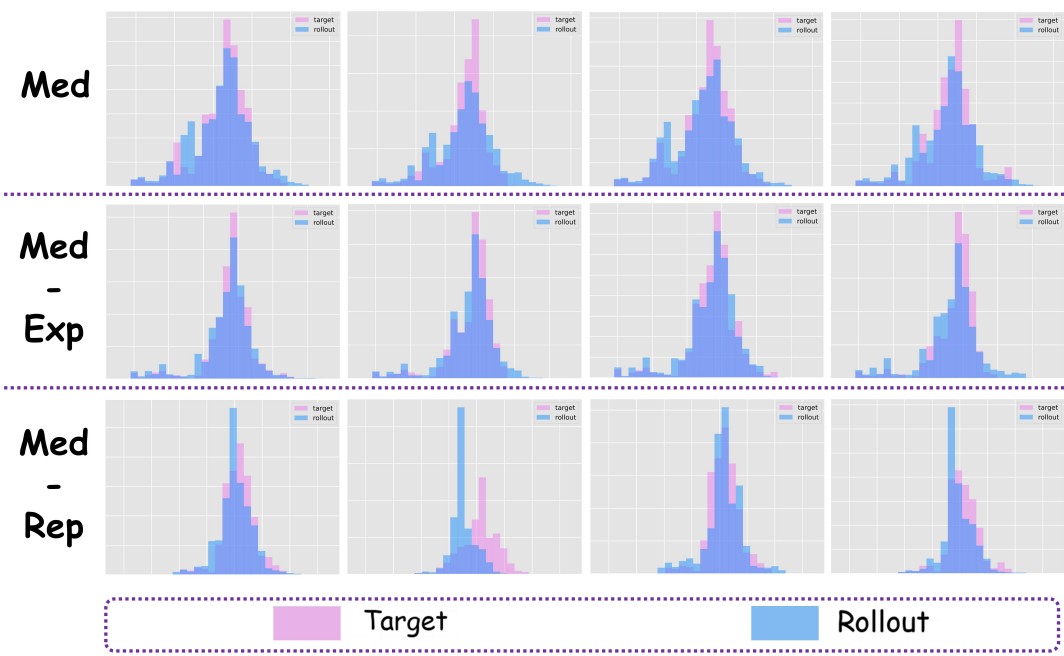

Figure 8: Within the **CEDC**, for the **Hopper** task, additional visualizations of distribution histograms across three datasets.

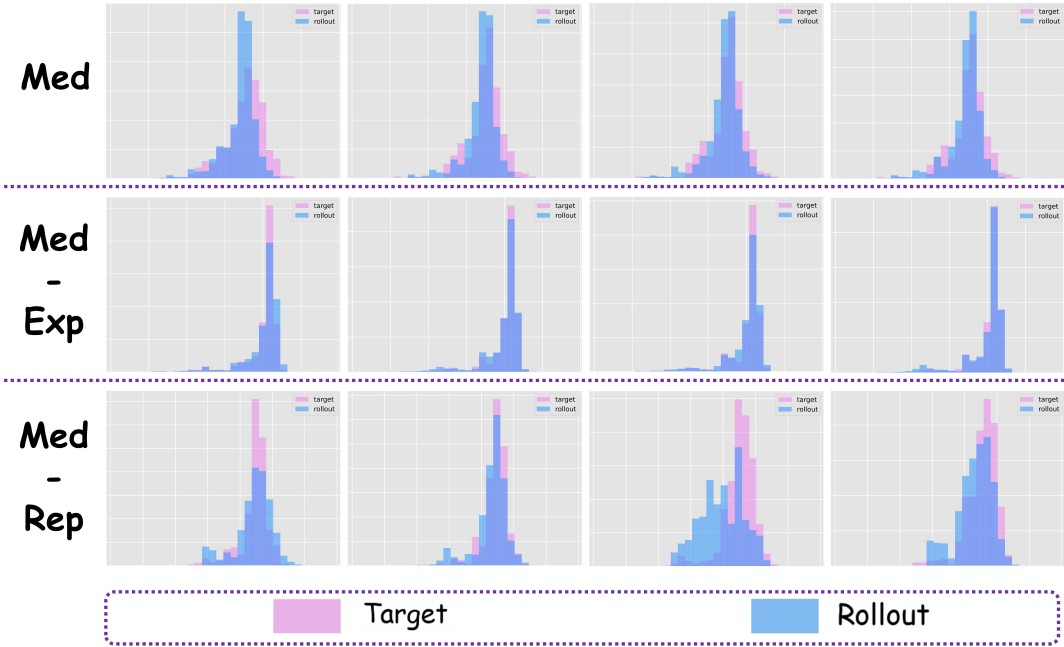

Figure 9: Within the **CEDC**, for the **Walker2d** task, additional visualizations of distribution histograms across three datasets.

**CEDMamba:** Figure 10 (HalfCheetah), Figure 11 (Hopper), Figure 12 (Walker2d).

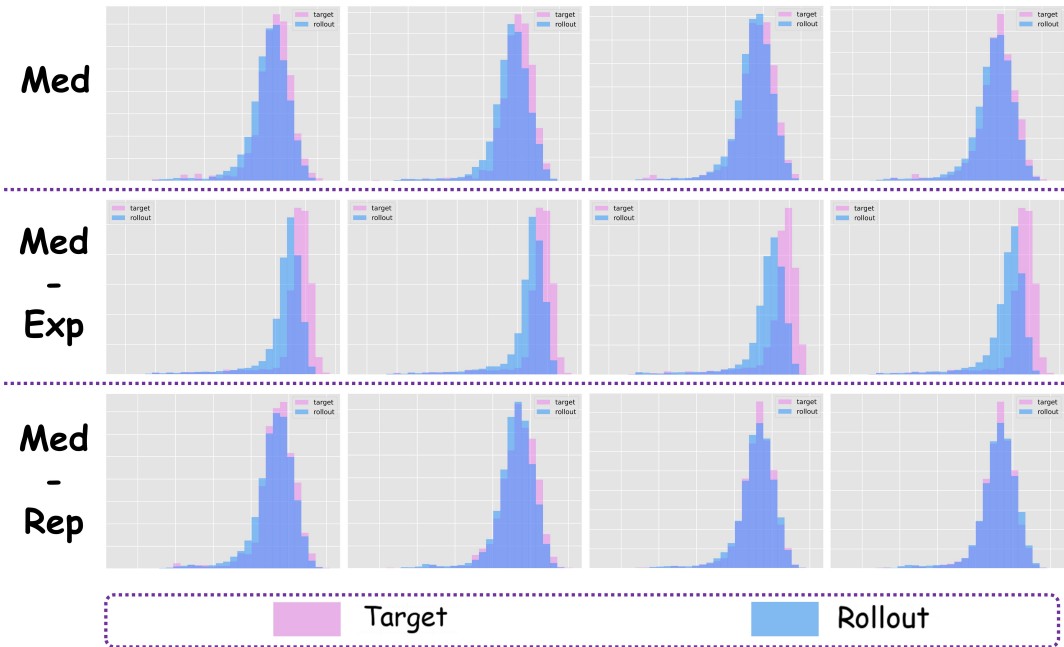

Figure 10: Within the **CEDMamba**, for the **HalfCheetah** task, additional visualizations of distribution histograms across three datasets.

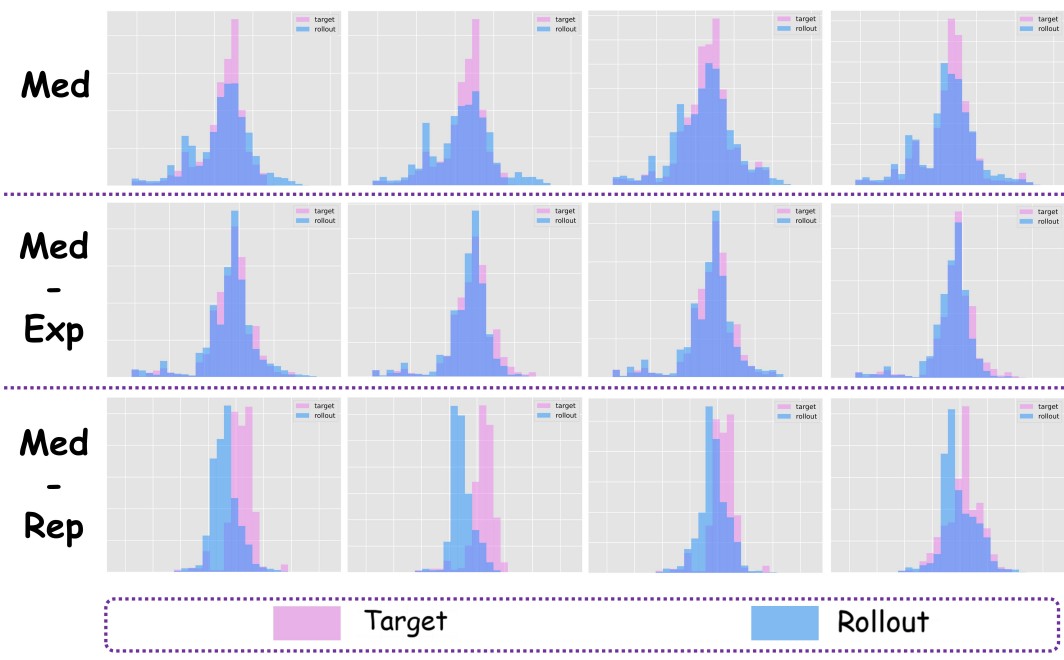

Figure 11: Within the **CEDMamba**, for the **Hopper** task, additional visualizations of distribution histograms across three datasets.

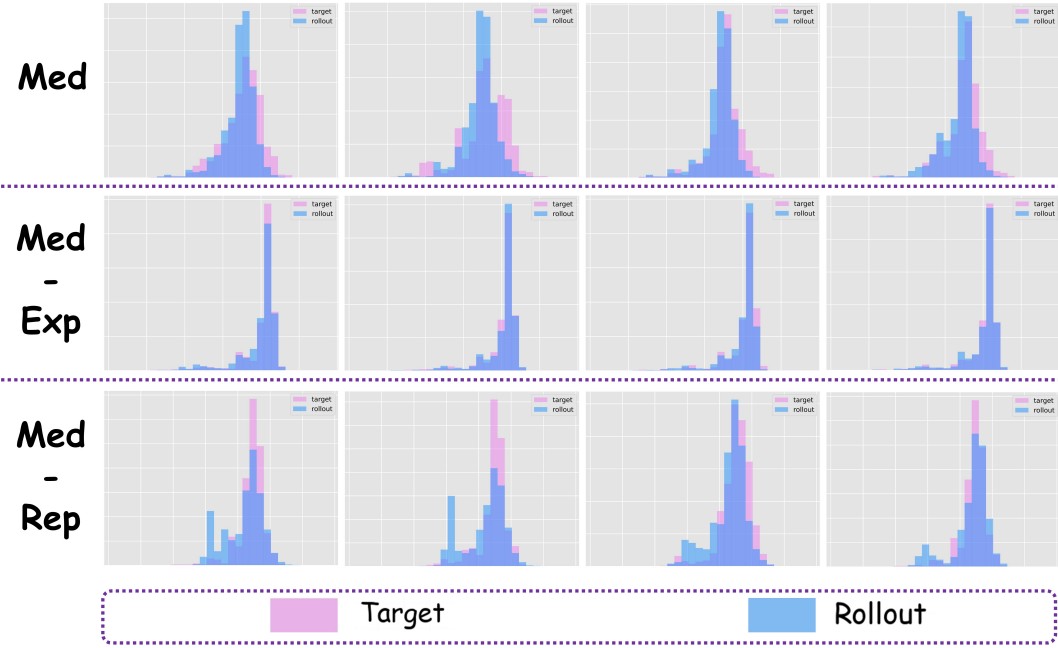

Figure 12: Within the **CEDMamba**, for the **Walker2d** task, additional visualizations of distribution histograms across three datasets.

## C  Details of Case Studies

### C.1  Dataset and Benchmark.

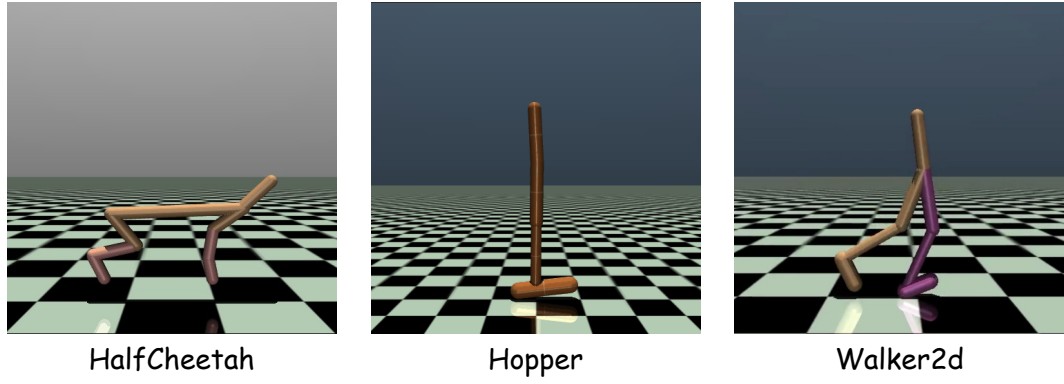

Figure 13: In our case studies, we focus on three specific tasks of the MuJoCo environment: HalfCheetah, Hopper, and Walker2d.

The MuJoCo environment tasks (HalfCheetah, Hopper, Walker2d) (Todorov et al., 2012) have been part of the D4RL benchmark (Fu et al., 2020), encompassing a series of continuous locomotion tasks characterized by dense reward signals. Specifically, the three tasks utilized in our case studies are defined as follows:

- **HalfCheetah:** The HalfCheetah is a two-dimensional robot consisting of nine interconnected body segments linked by eight joints, including a pair of paws. Its objective is to apply torques to these joints to drive the cheetah forward (to the right) as quickly as possible. Additionally, the HalfCheetah task has a 17-dimensional state space and a 6-dimensional action space.

- **Hopper:** The Hopper is a two-dimensional single-legged robot composed of four primary segments: the torso at the top, the thigh in the middle, the leg at the bottom, and a single foot on which the entire body rests. Its objective is to produce forward (rightward) locomotion by driving the three

joints linking these segments with appropriately applied torques, enabling it to perform continuous hopping motions. Furthermore, the Hopper task features an 11-dimensional state space and a 3-dimensional action space.

- **Walker2d:** The Walker is a two-dimensional bipedal robot composed of seven principal segments: a single upper torso from which the two legs diverge, a pair of thighs positioned below the torso, a pair of legs extending from the thighs, and two feet that provide ground support. The task is to make it walk forward (to the right) by applying torque to the six hinges linking these seven segments. Moreover, the Walker2d task has a 17-dimensional state space and a 6-dimensional action space.

For each of these, we utilize three different v2 datasets, each representing a different level of data quality: medium, medium-expert, and medium-replay. The medium dataset is obtained by first training a policy online with the Soft Actor-Critic algorithm, early-stopping the training process, and then collecting one million samples from the resulting partially trained policy. The medium-expert dataset is constructed by mixing equal amounts of expert demonstrations with suboptimal trajectories, where the suboptimal data is generated either from a partially trained policy or by unrolling a uniform-at-random policy. And the medium-replay dataset is created by recording every sample accumulated in the replay buffer over the course of training, up until the point at which the policy reaches the medium level of performance. Overall, MuJoCo offers an excellent testing ground to investigate how datasets, originating from policies with varying skill levels, influence learning and performance.

## C.2  Baselines.

Various domains, from computer vision (Sun et al., 2025a;b;c;d;e; Luo et al., 2025a;b; Yin et al., 2025; Fang et al., 2025) to natural language processing (Lv et al., 2025; Qin et al., 2025), increasingly leverage deep reinforcement learning (Xia et al., 2025) as a core paradigm for model training. Within this broad research landscape, trajectory optimization methods have emerged as a particularly promising direction. In our study, we adopt Decision Transformer, Decision ConvFormer, and Decision Mamba as representative baseline approaches.

**Decision Transformer (DT)** frames offline reinforcement learning as a sequence modeling problem, and leveraging the Transformer architecture. Instead of optimizing the value functions or the learning policies directly, DT treats the trajectories as sequences composed of states, actions, and returns-to-go (the sum of future rewards from each step). The agent is trained to predict the next action, conditioning on the previous states, actions, and desired return of the trajectory. At each timestep, input into the Transformer consists of a tuple: the current state, the previous action, and the target returns-to-go. It then autoregressively predicts the next action that should be taken to maximize the likelihood of achieving the return. Such an approach effectively turns policy generation into a sequence generation task, similar to how language models predict the next word in a sentence.

**Decision ConvFormer (DC)** serves as a novel offline reinforcement learning predictor that fundamentally rethinks how sequential dependencies are captured in offline reinforcement learning trajectories. While the traditional DT models offline reinforcement learning as a sequence modeling problem using the Transformer's self-attention mechanism, DT's reliance on attention can be suboptimal for offline RL tasks. Furthermore, DC replaces the global attention mechanism with local, causal 1D convolutional filters. Specifically, DC employs three independent convolutional mixers for states, actions, and returns, each operating over a local temporal window (e.g., 6 timesteps). This design ensures that predictions are made using only nearby tokens, effectively capturing the local, Markovian structure of RL data, while also dramatically reducing model complexity and computational cost.

**Decision Mamba (DMamba)** further replaces the self-attention mechanism in DT with the Mamba (Gu & Dao, 2024) block. The architecture comprises token-mixing and channel-mixing layers, with the Mamba block acting as the primary token-mixing module. By leveraging the structured state space modeling, DMamba potentially offering greater efficiency and the ability to model longer dependencies.

### C.3 Implementation Details.

To ensure the reproducibility of the experiment, we have included all codes in the supplementary materials. Herein, we encounter important hyperparameters and implementation details. All experiments are conducted on a single NVIDIA GeForce RTX 3090 graphics card. Hyperparameters are mainly set with reference to the Categorical Decision Transformer (Furuta et al., 2022). In Table 4, we list detailed hyperparameters of the CEDT. In Table 5, we list detailed hyperparameters of the CEDC. In Table 6, we list detailed hyperparameters of the CEDMamba.

Table 4: Hyperparameters for the CEDT.

| Hyperparameters | Value |
|---|---|
| Number of Bins | 31 |
| Number of Layers | 3 |
| Number of Attention Head | 1 |
| Embedding Dimension | 128 |
| Batch Size | 64 |
| Context Length | 20 |
| Dropout | 0.1 |
| Learning Rate | 1e-4 |
| Grad Norm Clip | 0.25 |
| Weight Decay | 1e-4 |
| Warmup Steps | 10000 |
| Activation Function | ReLU |
| Gamma | 1.0 |
| Max Training Iterations | 10 |
| Random Seeds | 0, 1, 2, 3, 4 |

Table 5: Hyperparameters for the CEDC.

| Hyperparameters | Value |
|---|---|
| Number of Bins | 31 |
| Number of Layers | 3 |
| Convolution Window Size | 6 |
| Embedding Dimension | 128 |
| Batch Size | 64 |
| Context Length | 20 |
| Dropout | 0.1 |
| Learning Rate | 1e-4 |
| Grad Norm Clip | 0.25 |
| Weight Decay | 1e-4 |
| Warmup Steps | 10000 |
| Activation Function | ReLU |
| Gamma | 1.0 |
| Max Training Iterations | 10 |
| Random Seeds | 0, 1, 2, 3, 4 |

Table 6: Hyperparameters for the CEDMamba.

| Hyperparameters | Value |
|---|---|
| Number of Bins | 31 |
| Number of Layers | 3 |
| Embedding Dimension | 128 |
| Batch Size | 64 |
| Context Length | 20 |
| Dropout | 0.1 |
| Learning Rate | 1e-4 |
| Grad Norm Clip | 0.25 |
| Weight Decay | 1e-4 |
| Warmup Steps | 10000 |
| Activation Function | ReLU |
| Gamma | 1.0 |
| Max Training Iterations | 10 |
| Random Seeds | 0, 1, 2, 3, 4 |

## D Evaluations Grounded in More Metrics

### D.1 An implementation of EACE

For a more thorough evaluation, this section introduces a tractable approximation. The EACE is defined as:

$$\text{EACE} = \mathbb{E}_{\hat{P}} \left[ \left| \mathbb{P}(\text{U} = 1 | \hat{P} = p) - p \right| \right]. \tag{30}$$

Overall, the primary bottleneck is that the true hidden policy $\pi$ is inaccessible, and it is difficult to ascertain whether the discrepancy between the two policies is smaller than $\delta$. Therefore, approximate solutions are adopted. Firstly, given that the policy $\pi$ is expressed in terms of its action outputs, we estimate it by performing rollouts to sample its behavior. Then, the accuracy is practically assessed according to:

$$\text{U} = \mathbb{I}(||a_{\text{pred}} - a_{\text{gt}}|| < \delta). \tag{31}$$

Furthermore, following standard practices in uncertainty estimation (Kendall & Gal, 2017; LeCun et al., 2006), we map the action prediction error to the confidence score through an exponential kernel:

$$\hat{p} = \exp(-\alpha ||a_{\text{pred}} - a_{\text{gt}}||^2). \tag{32}$$

Then, predictions are assigned to confidence bins $B_k = \{i | \hat{p}_i \in [b_k, b_{k+1})\}$, and within each bin, the mean confidence score and the corresponding actual accuracy are evaluated by:

$$\overline{p}_k = \frac{1}{|B_k|} \sum_{i \in B_k} \hat{p}_i \qquad \text{and} \qquad acc_k = \frac{1}{|B_k|} \sum_{i \in B_k} \text{U}_i. \tag{33}$$

Finally, the computation of EACE is conducted as:

$$\text{EACE} = \sum_{k=1}^{M} \frac{|B_k|}{N} \cdot |\overline{p}_k - acc_k|, \tag{34}$$

where $N$ is the number of samples. In the subsequent implementation, the hyperparameter settings for the evaluation are configured as: $\alpha = 10, \delta = 0.1$, and $N = 10$. The experimental results are presented in Table 7, Table 8 and Table 9.

Table 7: EACE comparison of CEDT and DT. **Bolded text** indicates that the results of CEDT performs as well as or better than DT.

| | CEDT | | | | | DT | | | | |
|---|---|---|---|---|---|---|---|---|---|---|
| | | | EACE ↓ | | | | | | | |
| | Med | Med-Exp | Med-Rep | Average | Total | Med | Med-Exp | Med-Rep | Average | Total |
| HC | **0.26±0.02** | **0.22±0.01** | **0.12±0.01** | **0.20** | | 0.26±0.01 | 0.27±0.01 | 0.13±0.01 | 0.22 | |
| Hp | 0.22±0.02 | **0.18±0.01** | 0.12±0.00 | 0.17 | **0.49** | 0.21±0.02 | 0.18±0.01 | 0.10±0.02 | 0.16 | 0.50 |
| Wk | **0.16±0.03** | 0.17±0.02 | **0.02±0.00** | **0.12** | | 0.16±0.03 | 0.16±0.02 | 0.03±0.00 | 0.12 | |

Table 8: EACE comparison of CEDC and DC. **Bolded text** indicates that the results of CEDC performs as well as or better than DC.

| | CEDC | | | | | DC | | | | |
|---|---|---|---|---|---|---|---|---|---|---|
| | | | EACE ↓ | | | | | | | |
| | Med | Med-Exp | Med-Rep | Average | Total | Med | Med-Exp | Med-Rep | Average | Total |
| HC | **0.12±0.01** | **0.03±0.00** | **0.03±0.01** | **0.06** | | 0.13±0.01 | 0.10±0.03 | 0.04±0.01 | 0.09 | |
| Hp | **0.04±0.01** | **0.04±0.01** | **0.02±0.01** | **0.03** | **0.12** | 0.07±0.01 | 0.05±0.01 | 0.02±0.00 | 0.05 | 0.17 |
| Wk | **0.04±0.01** | **0.04±0.01** | **0.00±0.00** | **0.03** | | 0.05±0.02 | 0.05±0.01 | 0.00±0.00 | 0.03 | |

Table 9: EACE comparison of CEDMamba and DMamba. **Bolded text** indicates that results of CEDMamba performs as well as or better than DMamba.

| | CEDMamba | | | | | DMamba | | | | |
|---|---|---|---|---|---|---|---|---|---|---|
| | | | EACE ↓ | | | | | | | |
| | Med | Med-Exp | Med-Rep | Average | Total | Med | Med-Exp | Med-Rep | Average | Total |
| HC | **0.04±0.03** | **0.02±0.01** | **0.02±0.01** | **0.03** | | 0.08±0.01 | 0.05±0.04 | 0.02±0.01 | 0.05 | |
| Hp | **0.03±0.02** | 0.05±0.01 | **0.02±0.00** | **0.03** | **0.07** | 0.05±0.01 | 0.04±0.01 | 0.03±0.00 | 0.04 | 0.11 |
| Wk | **0.01±0.00** | **0.02±0.01** | **0.00±0.00** | **0.01** | | 0.04±0.01 | 0.03±0.02 | 0.00±0.00 | 0.02 | |

### D.2 KL Divergence

To further validate the effectiveness of our method, we introduce KL divergence in this section for additional evaluation. The experimental results are presented in Table 10, Table 11, Table 12.

Table 10: KL divergence comparison of CEDT and DT. **Bolded text** indicates that the results of CEDT performs as well as or better than DT.

| | CEDT | | | | | DT | | | | |
|---|---|---|---|---|---|---|---|---|---|---|
| | | | KL Divergence ↓ | | | | | | | |
| | Med | Med-Exp | Med-Rep | Average | Total | Med | Med-Exp | Med-Rep | Average | Total |
| HC | **0.04±0.01** | **0.19±0.13** | **0.09±0.01** | **0.11** | | 0.05±0.01 | 0.24±0.09 | 0.16±0.11 | 0.15 | |
| Hp | **0.10±0.02** | **0.10±0.03** | **0.40±0.38** | **0.20** | **0.65** | 0.14±0.02 | 0.15±0.08 | 0.88±0.41 | 0.39 | 1.04 |
| Wk | **0.18±0.06** | **0.04±0.01** | 0.81±0.70 | **0.34** | | 0.33±0.19 | 0.57±0.68 | 0.60±0.29 | 0.50 | |

Table 11: KL divergence comparison of CEDC and DC. **Bolded text** indicates that the results of CEDC performs as well as or better than DC.

| | CEDC | | | | | DC | | | | |
|---|---|---|---|---|---|---|---|---|---|---|
| | KL Divergence ↓ | | | | | | | | | |
| | Med | Med-Exp | Med-Rep | Average | Total | Med | Med-Exp | Med-Rep | Average | Total |
| HC | **0.04±0.01** | **0.19±0.27** | **0.03±0.01** | **0.09** | | 0.06±0.02 | 0.19±0.21 | 0.14±0.09 | 0.13 | |
| Hp | **0.11±0.03** | **0.08±0.03** | **0.38±0.36** | **0.19** | **0.48** | 0.17±0.02 | 0.12±0.05 | 0.89±0.26 | 0.39 | 1.02 |
| Wk | **0.21±0.14** | **0.04±0.01** | **0.35±0.13** | **0.20** | | 0.33±0.11 | 0.68±0.60 | 0.49±0.08 | 0.50 | |

Table 12: KL divergence comparison of CEDMamba and DMamba. **Bolded text** indicates that results of CEDMamba performs as well as or better than DMamba.

| | CEDMamba | | | | | DMamba | | | | |
|---|---|---|---|---|---|---|---|---|---|---|
| | KL Divergence ↓ | | | | | | | | | |
| | Med | Med-Exp | Med-Rep | Average | Total | Med | Med-Exp | Med-Rep | Average | Total |
| HC | **0.04±0.01** | 0.67±0.30 | **0.03±0.01** | 0.25 | | 0.05±0.01 | 0.46±0.22 | 0.15±0.07 | 0.22 | |
| Hp | **0.10±0.03** | **0.06±0.01** | **0.82±0.69** | **0.33** | **0.73** | 0.11±0.03 | 0.19±0.07 | 0.95±0.40 | 0.42 | 1.19 |
| Wk | **0.19±0.05** | **0.03±0.01** | **0.23±0.06** | **0.15** | | 0.35±0.17 | 0.95±0.64 | 0.36±0.11 | 0.55 | |

### D.3   TV Divergence

Furthermore, we also conduct additional evaluation with the TV divergence. The experimental results are shown in Table 13, Table 14 and Table 15.

Table 13: TV divergence comparison of CEDT and DT. **Bolded text** indicates that the results of CEDT performs as well as or better than DT.

| | CEDT | | | | | DT | | | | |
|---|---|---|---|---|---|---|---|---|---|---|
| | TV Divergence ↓ | | | | | | | | | |
| | Med | Med-Exp | Med-Rep | Average | Total | Med | Med-Exp | Med-Rep | Average | Total |
| HC | **0.10±0.02** | **0.25±0.09** | **0.13±0.01** | **0.16** | | 0.10±0.01 | 0.29±0.06 | 0.16±0.07 | 0.18 | |
| Hp | **0.18±0.02** | **0.15±0.03** | **0.31±0.14** | **0.21** | **0.65** | 0.19±0.03 | 0.19±0.07 | 0.53±0.13 | 0.30 | 0.82 |
| Wk | **0.24±0.05** | **0.10±0.02** | 0.49±0.18 | **0.28** | | 0.27±0.10 | 0.33±0.32 | 0.42±0.13 | 0.34 | |

Table 14: TV divergence comparison of CEDC and DC. **Bolded text** indicates that the results of CEDC performs as well as or better than DC.

| | CEDC | | | | | DC | | | | |
|---|---|---|---|---|---|---|---|---|---|---|
| | TV Divergence ↓ | | | | | | | | | |
| | Med | Med-Exp | Med-Rep | Average | Total | Med | Med-Exp | Med-Rep | Average | Total |
| HC | **0.10±0.01** | **0.19±0.11** | **0.13±0.03** | **0.12** | | 0.13±0.03 | 0.23±0.11 | 0.15±0.04 | 0.17 | |
| Hp | **0.17±0.03** | **0.15±0.03** | **0.31±0.14** | **0.21** | **0.51** | 0.22±0.02 | 0.17±0.04 | 0.51±0.09 | 0.30 | 0.82 |
| Wk | **0.22±0.08** | **0.09±0.04** | **0.24±0.10** | **0.18** | | 0.28±0.05 | 0.41±0.30 | 0.35±0.06 | 0.35 | |

Table 15: TV divergence comparison of CEDMamba and DMamba. **Bolded text** indicates that results of CEDMamba performs as well as or better than DMamba.

| | CEDMamba | | | | | DMamba | | | | |
|---|---|---|---|---|---|---|---|---|---|---|
| | TV Divergence ↓ | | | | | | | | | |
| | Med | Med-Exp | Med-Rep | Average | Total | Med | Med-Exp | Med-Rep | Average | Total |
| HC | **0.11±0.02** | 0.39±0.12 | **0.06±0.02** | **0.19** | | 0.12±0.02 | 0.35±0.09 | 0.17±0.08 | 0.21 | |
| Hp | **0.17±0.02** | **0.12±0.01** | **0.47±0.18** | **0.25** | **0.59** | 0.18±0.03 | 0.21±0.04 | 0.54±0.12 | 0.31 | 0.88 |
| Wk | **0.17±0.06** | **0.08±0.02** | **0.21±0.04** | **0.15** | | 0.27±0.06 | 0.53±0.16 | 0.29±0.11 | 0.36 | |

# E Evaluations for Different Number of Bins

To further validate the effectiveness of our proposed approach, this section examines the impact of different bin counts on the resulting W1_Dis, return, and EACE. All experiments were conducted over five random seeds. As the main text already includes the results for the setting with 30 bins, we omit this case in the tables below and present only the results for bin=15, bin=60, and bin=120. The experimental results averaged over five random seeds are presented in Table 16, Table 17, and Table 18, while the corresponding standard deviations are reported in Table 19, Table 20, and Table 21.

Table 16: Mean Evaluation of Different Numbers of Bins on CEDT.

| | | HalfCheetah | | | Hopper | | | Walker2d | | |
|---|---|---|---|---|---|---|---|---|---|---|
| | | Med | Med-Exp | Med-Rep | Med | Med-Exp | Med-Rep | Med | Med-Exp | Med-Rep |
| W1_Dis | bin=15 | 0.22 | 0.76 | 0.26 | 0.15 | 0.18 | 0.19 | 0.27 | 0.09 | 0.42 |
| | bin=60 | 0.22 | 0.74 | 0.24 | 0.16 | 0.11 | 0.16 | 0.24 | 0.09 | 1.12 |
| | bin=120 | 0.24 | 0.74 | 0.28 | 0.14 | 0.15 | 0.21 | 0.31 | 0.11 | 0.55 |
| Return | bin=15 | 42.82 | 86.65 | 39.83 | 56.53 | 70.38 | 67.61 | 76.63 | 107.57 | 38.99 |
| | bin=60 | 42.75 | 86.08 | 39.76 | 63.29 | 91.13 | 72.11 | 77.24 | 107.41 | 36.10 |
| | bin=120 | 42.91 | 86.08 | 39.95 | 62.91 | 67.67 | 73.35 | 75.29 | 107.00 | 39.42 |
| EACE | bin=15 | 0.27 | 0.24 | 0.13 | 0.20 | 0.17 | 0.10 | 0.15 | 0.16 | 0.02 |
| | bin=60 | 0.28 | 0.21 | 0.12 | 0.22 | 0.17 | 0.12 | 0.15 | 0.15 | 0.02 |
| | bin=120 | 0.27 | 0.22 | 0.19 | 0.21 | 0.16 | 0.11 | 0.16 | 0.15 | 0.02 |

Table 17: Mean Evaluation of Different Numbers of Bins on CEDC.

|  |  | HalfCheetah | | | Hopper | | | Walker2d | | |
|---|---|---|---|---|---|---|---|---|---|---|
|  |  | Med | Med-Exp | Med-Rep | Med | Med-Exp | Med-Rep | Med | Med-Exp | Med-Rep |
| W1_Dis | bin=15 | 0.24 | 0.63 | 0.15 | 0.11 | 0.09 | 0.27 | 0.27 | 0.10 | 0.33 |
| | bin=60 | 0.22 | 0.50 | 0.18 | 0.16 | 0.10 | 0.18 | 0.18 | 0.11 | 0.39 |
| | bin=120 | 0.23 | 0.40 | 0.15 | 0.13 | 0.10 | 0.18 | 0.18 | 0.08 | 0.50 |
| Return | bin=15 | 43.07 | 87.14 | 40.86 | 59.98 | 81.08 | 67.18 | 77.02 | 107.40 | 45.18 |
| | bin=60 | 42.99 | 87.97 | 40.57 | 62.74 | 83.75 | 80.76 | 76.49 | 106.95 | 47.19 |
| | bin=120 | 43.11 | 88.98 | 40.49 | 62.03 | 84.52 | 80.41 | 77.95 | 107.73 | 45.44 |
| EACE | bin=15 | 0.10 | 0.03 | 0.03 | 0.07 | 0.04 | 0.03 | 0.06 | 0.04 | 0.00 |
| | bin=60 | 0.10 | 0.05 | 0.04 | 0.05 | 0.05 | 0.02 | 0.04 | 0.04 | 0.00 |
| | bin=120 | 0.11 | 0.06 | 0.03 | 0.07 | 0.05 | 0.03 | 0.05 | 0.04 | 0.00 |

Table 18: Mean Evaluation of Different Numbers of Bins on CEDMamba.

|  |  | HalfCheetah | | | Hopper | | | Walker2d | | |
|---|---|---|---|---|---|---|---|---|---|---|
|  |  | Med | Med-Exp | Med-Rep | Med | Med-Exp | Med-Rep | Med | Med-Exp | Med-Rep |
| W1_Dis | bin=15 | 0.24 | 0.78 | 0.16 | 0.10 | 0.12 | 0.14 | 0.23 | 0.12 | 0.20 |
| | bin=60 | 0.21 | 0.76 | 0.13 | 0.13 | 0.13 | 0.40 | 0.23 | 0.09 | 0.59 |
| | bin=120 | 0.21 | 0.80 | 0.17 | 0.15 | 0.11 | 0.24 | 0.18 | 0.09 | 0.42 |
| Return | bin=15 | 42.96 | 86.85 | 40.73 | 64.56 | 79.67 | 77.83 | 78.96 | 106.68 | 53.49 |
| | bin=60 | 43.08 | 86.15 | 40.87 | 59.43 | 89.46 | 82.57 | 76.68 | 107.20 | 47.03 |
| | bin=120 | 43.01 | 85.97 | 40.76 | 60.99 | 95.48 | 73.52 | 77.35 | 107.28 | 56.70 |
| EACE | bin=15 | 0.03 | 0.01 | 0.01 | 0.03 | 0.03 | 0.02 | 0.01 | 0.01 | 0.00 |
| | bin=60 | 0.04 | 0.01 | 0.03 | 0.04 | 0.04 | 0.02 | 0.02 | 0.01 | 0.00 |
| | bin=120 | 0.07 | 0.01 | 0.03 | 0.03 | 0.04 | 0.02 | 0.02 | 0.02 | 0.00 |

Table 19: Standard Derivation Evaluation of Different Numbers of Bins on CEDT.

|  |  | HalfCheetah | | | Hopper | | | Walker2d | | |
|---|---|---|---|---|---|---|---|---|---|---|
|  |  | Med | Med-Exp | Med-Rep | Med | Med-Exp | Med-Rep | Med | Med-Exp | Med-Rep |
| W1_Dis | bin=15 | 0.04 | 0.16 | 0.04 | 0.04 | 0.06 | 0.10 | 0.10 | 0.04 | 0.09 |
| | bin=60 | 0.05 | 0.04 | 0.07 | 0.03 | 0.02 | 0.04 | 0.10 | 0.04 | 1.02 |
| | bin=120 | 0.03 | 0.18 | 0.03 | 0.02 | 0.05 | 0.10 | 0.04 | 0.04 | 0.16 |
| Return | bin=15 | 0.24 | 1.25 | 0.17 | 1.51 | 14.79 | 16.91 | 1.44 | 0.56 | 16.55 |
| | bin=60 | 0.10 | 0.75 | 0.55 | 2.72 | 11.36 | 13.31 | 1.97 | 1.09 | 15.12 |
| | bin=120 | 0.17 | 1.67 | 0.20 | 4.22 | 15.57 | 10.51 | 3.44 | 0.85 | 7.50 |
| EACE | bin=15 | 0.02 | 0.01 | 0.01 | 0.02 | 0.01 | 0.01 | 0.02 | 0.02 | 0.00 |
| | bin=60 | 0.04 | 0.04 | 0.01 | 0.03 | 0.02 | 0.01 | 0.02 | 0.03 | 0.01 |
| | bin=120 | 0.02 | 0.02 | 0.01 | 0.02 | 0.01 | 0.01 | 0.02 | 0.02 | 0.00 |

Table 20: Standard Derivation Evaluation of Different Numbers of Bins on CEDC.

| | | HalfCheetah | | | Hopper | | | Walker2d | | |
|---|---|---|---|---|---|---|---|---|---|---|
| | | Med | Med-Exp | Med-Rep | Med | Med-Exp | Med-Rep | Med | Med-Exp | Med-Rep |
| W1_Dis | bin=15 | 0.05 | 0.30 | 0.05 | 0.05 | 0.03 | 0.22 | 0.09 | 0.04 | 0.15 |
| | bin=60 | 0.04 | 0.32 | 0.07 | 0.04 | 0.03 | 0.07 | 0.07 | 0.07 | 0.13 |
| | bin=120 | 0.06 | 0.13 | 0.07 | 0.02 | 0.03 | 0.09 | 0.03 | 0.03 | 0.19 |
| Return | bin=15 | 0.17 | 2.12 | 0.31 | 3.33 | 19.64 | 12.74 | 5.21 | 0.89 | 2.80 |
| | bin=60 | 0.21 | 2.44 | 0.36 | 3.79 | 5.15 | 8.55 | 1.78 | 1.55 | 9.92 |
| | bin=120 | 0.20 | 1.41 | 0.77 | 1.67 | 14.08 | 12.02 | 0.99 | 0.55 | 7.33 |
| EACE | bin=15 | 0.04 | 0.01 | 0.01 | 0.02 | 0.01 | 0.00 | 0.02 | 0.02 | 0.00 |
| | bin=60 | 0.01 | 0.02 | 0.02 | 0.01 | 0.01 | 0.00 | 0.02 | 0.01 | 0.00 |
| | bin=120 | 0.03 | 0.03 | 0.01 | 0.02 | 0.01 | 0.00 | 0.02 | 0.01 | 0.00 |

Table 21: Standard Derivation Evaluation of Different Numbers of Bins on CEDMamba.

| | | HalfCheetah | | | Hopper | | | Walker2d | | |
|---|---|---|---|---|---|---|---|---|---|---|
| | | Med | Med-Exp | Med-Rep | Med | Med-Exp | Med-Rep | Med | Med-Exp | Med-Rep |
| W1_Dis | bin=15 | 0.03 | 0.27 | 0.02 | 0.04 | 0.05 | 0.09 | 0.02 | 0.04 | 0.10 |
| | bin=60 | 0.06 | 0.16 | 0.05 | 0.03 | 0.05 | 0.19 | 0.06 | 0.03 | 0.28 |
| | bin=120 | 0.05 | 0.22 | 0.07 | 0.05 | 0.02 | 0.10 | 0.03 | 0.02 | 0.08 |
| Return | bin=15 | 0.24 | 1.64 | 0.22 | 4.23 | 13.83 | 11.52 | 0.40 | 0.90 | 10.93 |
| | bin=60 | 0.11 | 1.52 | 0.26 | 3.00 | 16.37 | 5.40 | 2.31 | 0.85 | 5.46 |
| | bin=120 | 0.18 | 1.91 | 0.45 | 3.97 | 9.71 | 15.42 | 3.55 | 0.48 | 13.50 |
| EACE | bin=15 | 0.02 | 0.01 | 0.01 | 0.02 | 0.01 | 0.01 | 0.02 | 0.02 | 0.00 |
| | bin=60 | 0.04 | 0.04 | 0.01 | 0.03 | 0.02 | 0.01 | 0.02 | 0.03 | 0.01 |
| | bin=120 | 0.02 | 0.02 | 0.01 | 0.02 | 0.01 | 0.01 | 0.02 | 0.02 | 0.00 |

# F   Evaluations for Different Size of Datasets

In this section, we investigate how different dataset sizes influence performance of the method. Specifically, we construct subsets by sampling 20%, 40%, and 80% of the trajectories from the *medium* datasets in each environment. We abandon using medium-expert and medium-replay for this analysis, as the trajectories in these datasets exhibit large inter-trajectory discrepancies that could confound the evaluation.

Table 22: Evaluation of Different Size of Datasets on CEDT and DT.

|  |  | CEDT | | | DT | | |
|---|---|---|---|---|---|---|---|
|  |  | HalfCheetah | Hopper | Walker2d | HalfCheetah | Hopper | Walker2d |
| W1_Dis | 20% | 0.24±0.06 | 0.13±0.04 | 0.23±0.02 | 0.25±0.04 | 0.20±0.06 | 0.23±0.06 |
|  | 40% | 0.19±0.04 | 0.13±0.06 | 0.32±0.06 | 0.20±0.07 | 0.14±0.03 | 0.31±0.06 |
|  | 80% | 0.19±0.04 | 0.12±0.01 | 0.26±0.08 | 0.25±0.03 | 0.16±0.03 | 0.34±0.12 |
| Return | 20% | 42.87±0.49 | 60.91±3.21 | 76.35±2.51 | 42.50±0.59 | 55.22±6.24 | 77.55±2.43 |
|  | 40% | 42.83±0.59 | 60.84±6.24 | 74.20±2.43 | 42.78±0.39 | 59.20±5.66 | 76.80±1.97 |
|  | 80% | 42.84±0.23 | 59.05±2.07 | 76.37±3.51 | 42.85±0.17 | 57.74±4.08 | 75.69±2.35 |

Table 23: Evaluation of Different Size of Datasets on CEDC and DC.

|  |  | CEDC | | | DC | | |
|---|---|---|---|---|---|---|---|
|  |  | HalfCheetah | Hopper | Walker2d | HalfCheetah | Hopper | Walker2d |
| W1_Dis | 20% | 0.25±0.08 | 0.13±0.02 | 0.23±0.04 | 0.23±0.04 | 0.19±0.03 | 0.26±0.04 |
|  | 40% | 0.21±0.06 | 0.13±0.03 | 0.26±0.06 | 0.23±0.03 | 0.15±0.04 | 0.38±0.15 |
|  | 80% | 0.21±0.11 | 0.13±0.03 | 0.25±0.05 | 0.24±0.07 | 0.14±0.04 | 0.37±0.10 |
| Return | 20% | 42.90±0.39 | 63.95±5.49 | 79.10±0.31 | 43.00±0.30 | 58.69±2.24 | 77.79±0.83 |
|  | 40% | 42.95±0.27 | 63.34±5.64 | 77.89±1.90 | 43.11±0.29 | 60.95±1.62 | 77.32±2.53 |
|  | 80% | 42.95±0.11 | 66.32±4.66 | 77.99±3.03 | 43.07±0.27 | 62.17±5.03 | 78.62±2.25 |

Table 24: Evaluation of Different Size of Datasets on CEDMamba and DMamba.

|  |  | CEDMamba | | | DMamba | | |
|---|---|---|---|---|---|---|---|
|  |  | HalfCheetah | Hopper | Walker2d | HalfCheetah | Hopper | Walker2d |
| W1_Dis | 20% | 0.23±0.07 | 0.15±0.05 | 0.26±0.05 | 0.24±0.04 | 0.15±0.03 | 0.38±0.15 |
|  | 40% | 0.18±0.01 | 0.14±0.05 | 0.22±0.07 | 0.24±0.03 | 0.11±0.03 | 0.36±0.07 |
|  | 80% | 0.27±0.03 | 0.12±0.03 | 0.24±0.08 | 0.23±0.05 | 0.12±0.03 | 0.56±0.52 |
| Return | 20% | 42.81±0.33 | 59.74±6.33 | 78.31±1.83 | 42.80±0.37 | 65.77±4.84 | 75.49±2.71 |
|  | 40% | 42.95±0.23 | 61.81±4.71 | 77.64±0.64 | 42.81±0.25 | 69.69±2.71 | 75.73±1.55 |
|  | 80% | 43.13±0.26 | 61.90±3.09 | 78.23±1.72 | 43.22±0.16 | 72.10±8.20 | 68.78±14.10 |

# G  Comparison with Cal-QL and Other Q-learning Methods

To further demonstrate the effectiveness of CEDM, we include comparisons against the offline part of Cal-QL (Nakamoto et al., 2024), BCQ (Fujimoto et al., 2019), and CQL (Kumar et al., 2020). Specifically, results of Cal-QL are sourced from Table 5 in its original publication, whereas the results for BCQ and CQL are obtained from the Table 2 of D4RL (Fu et al., 2020).

Table 25: Comparison of Return with Cal-QL and Other Q-learning Methods (with Total).

| | HalfCheetah | | | Hopper | | | Walker2d | | | Total |
|---|---|---|---|---|---|---|---|---|---|---|
| | Med | Med-Exp | Med-Rep | Med | Med-Exp | Med-Rep | Med | Med-Exp | Med-Rep | |
| CEDT | 42.77 | 86.64 | 39.59 | 59.85 | 78.88 | 62.11 | 73.77 | 107.70 | 37.48 | 588.79 |
| CEDC | 42.92 | 88.15 | 41.17 | 62.67 | 75.06 | 75.14 | 76.09 | 108.05 | 44.64 | 613.89 |
| CEDMamba | 42.87 | 85.59 | 40.86 | 59.73 | 90.28 | 78.19 | 76.69 | 108.05 | 56.56 | 638.82 |
| DT | 42.88 | 84.05 | 39.09 | 57.80 | 90.16 | 20.34 | 72.87 | 107.49 | 31.66 | 546.34 |
| DC | 42.81 | 88.16 | 39.69 | 59.13 | 76.57 | 30.84 | 74.87 | 104.36 | 50.81 | 567.24 |
| DMamba | 42.97 | 86.73 | 39.26 | 69.39 | 97.47 | 41.03 | 70.47 | 87.20 | 55.63 | 590.15 |
| Cal-QL | 52 | 54 | 51 | 89 | 69 | 76 | 75 | 96 | 52 | 614 |
| BCQ | 40.7 | 64.7 | 38.2 | 54.5 | 110.9 | 33.1 | 53.1 | 57.5 | 15.0 | 467.7 |
| CQL | 44.4 | 62.4 | 46.2 | 58.0 | 98.7 | 48.6 | 79.2 | 111.0 | 26.7 | 575.2 |

