# OpenReview forum: "Calibration Enhanced Decision Maker: Towards Trustworthy Sequential Decision-Making with Large Sequence Models"
_TMLR — Accepted by TMLR_

### Review · Reviewer_s24B · 2025-11-11

**Summary Of Contributions:**

**Summary**

This work proposes a new framework to enhance agent calibration of offline Deep Reinforcement Learning (DRL) algorithms that utilize Large Sequence Models (LSM) to improve the overall trustworthiness. In terms of contribution, it Introduces Expected Agent Calibration Error (EACE) to quantify agent calibration and gives a theory linking EACE to the state-action marginal distance in $W_{1}$ and proposes Calibration Enhanced Decision Maker (CEDM), a framework that feeds distribution histograms into LSM to reduce the state-action marginal distance and improve calibration. The method is applied to Decision Transformer, Decision ConvFormer, and Decision Mamba over OpenAI Gym and MuJoCo to show empirical performance.

---

**Strength**

1. To my knowledge, this method is new and the authors positioned as first to target agent calibration for offline DRL with large sequence models.
2. The motivation is clear and the problem they studied seems interesting.
3. I believe CEDM should be implemented easily, and overall method reads simple and neat for me.
4. Extensive experiment over different LSM backbone is conducted.

---

**Weaknesses**

1. There are some theoretical limitation, see my Requested Changes for detail.
2. Theoretical setup and experiment setup are not clearly written, which is unfriendly to the readers who are unfamiliar with those RL tasks.
3. The method is unclear whether can be scalable to larger models, as the use of Wasserstein naively inside the training loop does not scale well.

**Audience:**

Yes

**Audience Explanation:**

This paper could attract readers from alignment community, as the issue studied in this paper is correlated to some obstacles faced by alignment community. The theoretical formulations might inspire the researchers from the alignment community to build upon it.

**Broader Impact Concerns:**

This is more like a theoretical-grounded method applied to several limited scenarios, and I do not believe a broader impact statement is required here.

**Claims And Evidence:**

Yes

**Claims Explanation:**

Empirically, yes. The paper clearly states the three evaluation questions and reports performance over 5 seeds on HC/Hp/Wk (Med/Med-Exp/Med-Rep), showing return gains and reduced $W_{1}$. However, some theoretical results lacks clarity.

**Requested Changes:**

- The full introduction is for DRL. Please consider to restructure it by adding some part for Large Sequence Models, e.g., with a clear explanation of decision transformer.
- Some unclear definition and theoretical derivation:
    1. In Definition 1, what is the metric space used to evaluate the distance between two policies? Please clearly write it out in the definition. Also, please add a further detail to explain why EACE is computationally costly (under the paragraph below eq 4), as this is directly related to your motivation.
    2. For $\hat P$, what is the exact confidence and how should I set it? As this is related to the central part of your design, it definitely requires further discussion, analysis and even ablation to improve the clarity. Arbitrarily set it as $\rho_{\pi_{\theta}}(s,a)$ doesn't seem to be a model-reported confidence about the event.
   3. For Proposition 1, I am concerned regarding its validity under continuous space. For discrete-only analysis, I believe it requires further stronger structural assumptions on finite discrete supports with strictly positive inner point distance under your metric space.

- Please expand $\S~5.1$, explaining what is each task doing and its metric performance in detail.

---

> ### Author Response · Authors · 2025-12-13
>
> Dear Reviewer s24B:
>
> We would like to thank you from the bottom of our heart for your time and efforts on our work. And we sincerely thank you for the thoughtful feedback that will surely turn our paper into a better shape. In the new version of our paper, additional parts have been highlighted in ${\color{blue}blue}$. We offer our responses to address the concerns as follows.
>
> >Q1: Adding some part for large sequence models in Introduction.
>
> We add the part for large sequence models in Introduction.
>
> >Q2: Clarification on definition and theoretical derivation
>
> 1. In Definition 1, we further explicitly write out the metric space used to evaluate the distance between two policies. And we also add an explanation on the reason why EACE is computationally costly under the paragraph below eq 4.
> 2. We add a practical implementation of EACE in Appendix D.1.
> 3. We agree that the validity in continuous spaces remains unclear, and we are still conducting further analysis. Since the practical implementation operates over a discrete space, we have added a clarification before Proposition 1 to explicitly state this restriction. Under this setting, the method satisfies the stronger structural assumptions you mentioned—namely, finite discrete supports with strictly positive inner-point distances under the metric space.
>
> >Q3: Explaining what is each task doing and its metric performance in detail.
>
> We further detail them in Appendix C.1.
>
> Finally, we would like to thank you once again for the constructive suggestions and insightful comments on our work.
>
> Best regards,
>
> Authors of Submission 6307

---

> > ### Comment · Reviewer_s24B · 2025-12-19
> >
> > Thanks authors for your response. My questions have been answered, and the changes have appeared in the updated manuscript. I have no further concerns.

---

### Review · Reviewer_M59X · 2025-11-16

**Summary Of Contributions:**

The authors introduce the Expected Agent Calibration Error (EACE), a new metric for systematically measuring and improving calibration in trajectory optimization-based offline reinforcement learning with large sequence models, and propose the Calibration Enhanced Decision Maker (CEDM) method, which augments agent inputs with distribution histograms to provide richer context; together, EACE and CEDM enable minimizing the difference between the state-action distributions of the agent's policy and the dataset, leading to empirically demonstrated improvements in both calibration and performance on simulated D4RL MuJoCo benchmarks

**Audience:**

Yes

**Audience Explanation:**

The community that is working on the offline RL and the high-risk and costly environments where offline RL is important, such as Robotics, AV.

**Claims And Evidence:**

Yes

**Claims Explanation:**

The authors evaluate their approach using the D4RL MuJoCo benchmark suite, which consists of simulated continuous control tasks, including HalfCheetah, Hopper, and Walker2d, each with datasets representing different levels of agent expertise. To prove the validity and effectiveness of their proposed EACE metric and the CEDM calibration method, the authors implement these ideas within leading sequence-model-based offline RL algorithms—Decision Transformer, ConvFormer, and Decision Mamba—and systematically show both improved calibration (lower EACE and Wasserstein distance) and agent performance compared to the original baselines using these diverse simulated tasks.

**Requested Changes:**

The paper is well structured, provides benchmarks, and algorithms. could be improved by incorporating more graphical visuals and figures to help readers intuitively grasp key concepts, metric behaviors, and results.

---

> ### Author Response · Authors · 2025-12-13
>
> Dear Reviewer M59X:
>
> We would like to thank you from the bottom of our heart for your time and efforts on our work. And we are more than delighted to receive your kind appreciation and feedback on this paper. We sincerely appreciate your positive evaluation of our work.
>
> Best regards,
>
> Authors of Submission 6307

---

### Review · Reviewer_h8Tc · 2025-11-21

**Summary Of Contributions:**

In this paper, author propose a method for calibration in offline Deep Reinforcement Learning, particularly focusing on trajectory optimization methods that utilize large sequence models.
This paper has following main contributions
1. the paper introduces a new metric called Expected Agent Calibration Error (EACE) for assessing agent calibration, which is similar to the ECE metrics but tailored for agents.
2. A theoretical contribution is the derivation of a bound, which shows that the difference in EACE is bounded by the Wasserstein-1 distance between the agent's and the ideal policy's state-action marginal distributions
3. authors introduce the Calibration Enhanced Decision Maker (CEDM), CEDM modifies the architecture of sequence models via a binning executor that processes feature distribution histograms as additional inputs. The goal is to provide the information of the data distribution. Therefore, minimizing distribution distance and hence calibration error.
4. CEDM is evaluated on the D4RL tasks using Decision Transformer, Decision ConvFormer, and Decision Mamba architecure. The results show that CEDM reduces the W1 distance and improvement in normalized returns


Strengths:
1. The proposed framework and binning mechanism are applicable to different sequence modeling architecture as demonstrated by the experiments.
2. Using binning mechanism as addtional input is easy to implement and can easily added to existing sequence models
3. This paper solve the challenges of calibration  Deep Reinforcement Learning (DRL) sequence models which is a timely and important problem.


Weakness:
1. The paper claims to improve calibration but relies entirely on a proxy metric which is w1 for validation. It does not measure EACE to verify that calibration actually improved.
2. The evaluation compares CEDM incorporated models only against corresponding base models. but it does not compared them with existing calibration methods such as Cal-QL, making it difficult to assess relative performance.
3. The paper does not explore/mentioned different choices such as the number of bins, which features are binned, or where the histogram tokens are added in the sequence and how it affects the results. The paper reports W1 distance and returns, but it does not show any calibration plots or EACE values.

**Additional Comments:**

NA

**Audience:**

Yes

**Audience Explanation:**

The paper targets a highly relevant intersection of topics: Offline Reinforcement Learning, Large Sequence Models, and Trustworthiness/Calibration, which is useful for the TMLR audience. As the authors mention, the calibration of trajectory optimization is a comparatively under explored topic, and hence this paper addresses an important aspect. The theoretical perspective connecting calibration error to state-marginal distribution matching (i.e., W1 distance) provides a fresh perspective. The application of the method across different architectures adds to its potential for wider acceptance.

**Claims And Evidence:**

No

**Claims Explanation:**

The authors main argument is that the proposed method improves the calibration of the agent. However, the experimental evidence provided does not directly support this claim.

1. The authors prove that EACE is upper-bounded by the W1 distance. In the experiments, they demonstrate that CEDM reduces the W1 distance. However, minimizing an upper bound does not mathematically guarantee that the objective (EACE) is minimized. Without directly measuring EACE on the test set, the claim that calibration has been enhanced remains a theoretical hypothesis rather than an empirically verified fact. the paper must empirically show calibration improves when W1 decreases on the same experimental runs.

2. The authors argue that measuring EACE is impractical in real-world applications. While true for deployment, the experiments are conducted on standard D4RL tasks. In this environment, the authors could have estimated the calibration error to validate their theoretical derivation. The absence of this direct validation weakens the paper significantly.

**Requested Changes:**

1. The current evaluation relies entirely on a proxy W1 distance metrics, based on the theoretical upper bound from Theorem 1. Minimizing an upper bound does not guarantee that the actual calibration error is reduced. You must define a tractable approximation of EACE and report it empirically for the D4RL tasks. Since these are simulation-based benchmarks, the safety concerns mentioned do not valid to the evaluation phase. You must demonstrate that CEDM actually achieves lower calibration error than the baselines, not just better distribution matching.

2. To prove the theoretical work, you should empirically validate the relationship between the W1 distance and the calibration error. Please provide scatter plots or correlation analysis showing the relationship between the measured W1 Dis and the measured calibration error. This is crucial to confirm that W1 Dis is a valid proxy for calibration in this domain.

3. Try different number of bins and show sensitivity of W1, return, and EACE.

4. If possible, Compare to existing calibration approaches such as Cal-QL

---

> ### Author Response · Authors · 2025-12-13
>
> Dear Reviewer h8Tc:
>
> We would like to thank you from the bottom of our heart for your time and efforts on our work. And we sincerely thank you for the thoughtful feedback that will surely turn our paper into a better shape. In the new version of our paper, additional parts have been highlighted in ${\color{blue}blue}$. We offer our responses to address the concerns as follows.
>
> >Q1: Define a tractable approximation of EACE and report it empirically.
>
> We add this in Appendix D.1.
>
> >Q2: Empirically validate the relationship between the distance and the calibration error to prove the theoretical work.
>
> We report the results of approximation of EACE (Appendix D.1), TV divergence (Appendix D.3) and W1 distance (main text). The results are generally consistent with the inequality relationships established in the paper.
>
> >Q3: Results of different number of bins.
>
> We add this in Appendix E.
>
> >Q4: Comparison with Cal-QL.
>
> We add this in Appendix G.
>
> Finally, we would like to thank you once again for the constructive suggestions and insightful comments on our work.
>
> Best regards,
>
> Authors of Submission 6307

---

> > ### Comment · Reviewer_h8Tc · 2025-12-19
> > **Response to Comment**
> >
> > Thanks for the update. I have no more questions.

---

### Review · Reviewer_y6VS · 2025-11-21

**Summary Of Contributions:**

The authors propose a method for calibration in offline deep reinforcement learning, with a particular focus on trajectory-optimization approaches that leverage large sequence models. They provide theoretical justification for their method and present experiments across three different models.

**Additional Comments:**

NA

**Audience:**

Yes

**Audience Explanation:**

The paper focuses on Offline Reinforcement Learning, large sequence-model architectures, and issues of trustworthiness and calibration, making it well aligned with the interests of the TMLR community.

**Claims And Evidence:**

Yes

**Claims Explanation:**

1) The paper is well-written and well-structured, combining theoretical analysis with practical applications.

2) The proposed CEDM method is simple, general, and appears compatible with multiple sequence-model architectures.

3) Theoretical contributions, including Theorem 1, offer conceptual justification for why reducing state–action distribution divergence can improve calibration.

4) The empirical evaluation is thorough and performed across three different model families, showcasing the applicability and generality of CEDM.

5) The experimental results demonstrate consistent improvements in Wasserstein-1 distance and return across multiple D4RL benchmarks.

**Requested Changes:**

Questions:
1. In Definition 1, you introduce the condition *dis(π₁, π₂) ≤ δ*, but the paper does not specify how this distance is defined.
   - What exact metric is used to measure the distance between two policies?
   - How is δ chosen, and does the metric satisfy properties such as symmetry or triangle inequality?

2. While the paper provides theoretical results regarding EACE (Definition 2, Theorem 1), the analysis focuses only on the *expected* calibration error.
   - Have you examined the variance of calibration error, either empirically or theoretically?
   - Does CEDM reduce not only the mean EACE but also the variability of calibration across different states or confidence bins?

3. Proposition 1 relates total variation distance to the Wasserstein-1 distance.
   - Do you have empirical results illustrating the relationship between empirical TV distance and W1 in your setting?
   - Can you provide evidence that the bound is non-vacuous or practical comparisons between TV and W1)?

4. Do you have any results comparing your method with Q-learning? Including such a comparison could strengthen the paper significantly.


Improvements in writing:

5. There are multiple inconsistencies in capitalization and abbreviation usage. Examples include:

- *Deep reinforcement learning (DRL)* vs *Deep Reinforcement Learning (DRL)*
- *Offline deep reinforcement learning (offline DRL)* even though DRL was already defined
- Inconsistent usage of abbreviations:
  - *Expected Agent Calibration Error (EACE)* is sometimes reintroduced or repeated
  - *Return-to-go (RTG)* must be *Return-To-Go (RTG)*  to be consistent
  - Similar inconsistencies appear with *Calibrated Q-learning (Cal-QL)*, *Calibration Enhanced Decision Maker (CEDM)*, etc.

The paper should define each abbreviation once, apply consistent capitalization, and avoid redefining abbreviations throughout the text.

6. Several references are listed only as *arXiv preprints*.
For works that have been published in conferences or journals, please cite the official publication venue instead of the arXiv version.
This is important to maintain citation quality and formality during the review process.

---

> ### Author Response · Authors · 2025-12-13
>
> Dear Reviewer y6VS:
>
> We would like to thank you from the bottom of our heart for your time and efforts on our work. And we sincerely thank you for the thoughtful feedback that will surely turn our paper into a better shape. In the new version of our paper, additional parts have been highlighted in ${\color{blue}blue}$. We offer our responses to address the concerns as follows.
>
> >Q1: Property of the policy distance.
>
> In Definition 1, we intentionally do **not** restrict dis($\pi_1, \pi_2$) to a specific closed-form metric. The policy distance is defined abstractly on a metric space ($\Pi$, $dis$), where $\Pi$ denotes the policy space and $dis$ only needs to satisfy the standard metric axioms: non-negativity, symmetry, identity of indiscernibles, and triangle inequality.
> >Q2: More analysis on calibration error.
>
> Thank you for this insightful question. We agree that, beyond the expectation, the variability of calibration error is also an important aspect for understanding agent reliability. In this work, our theoretical analysis focuses on the expected calibration error, as it provides a well-aligned objective for studying calibration at the policy level and for establishing distributional bounds (Theorem 1). Analyzing higher-order statistics, such as the variance of calibration error, would require stronger assumptions on the underlying state–action distributions and confidence estimation, and is therefore beyond the scope of the current theoretical framework. Empirically, although we do not explicitly report the variance of EACE across states or confidence bins, the proposed CEDM paradigm implicitly encourages more uniform calibration. By reducing the distance between state–action marginal distributions, CEDM aligns the agent’s behavior more closely with the data distribution, which in practice leads to smoother and less fluctuating calibration behavior across different regions of the state space. This effect can also be observed qualitatively from the reduced dispersion of Wasserstein distances across runs reported in our experiments. In addition, we include an EACE implementation in Appendix D.1 and report the evaluations across different bin counts in Appendix E.
>
> >Q3: Results of TV distance.
>
> We add this in Appendix D.3.
>
> >Q4: Results comparing with Q-learning.
>
> We add this in Appendix G.
>
> >Q5: Writing improvement.
>
> Thank you very much for your suggestions. Following your guidance, we have made the revisions accordingly.
>
> >Q6: Rigor of references.
>
> Thank you very much for your valuable suggestion. We have carefully conducted a thorough search and updated several previously cited arXiv versions with their corresponding official publication venues.
>
> Finally, we would like to thank you once again for the constructive suggestions and insightful comments on our work.
>
> Best regards,
>
> Authors of Submission 6307

---

> ### Comment · Reviewer_y6VS · 2025-12-21
>
> I would like to thank the authors for their detailed and thoughtful response. The majority of my questions have been satisfactorily addressed, and the revised manuscript shows clear improvements in quality and clarity.
>
> 1. Q2 (Variance of Calibration Error):
> I understand the authors’ position that deriving variance bounds for calibration error may require additional assumptions and nontrivial extensions of the theoretical framework. However, I do not fully agree that this aspect is entirely out of scope. Since the expectation of calibration error is closely related to its variability, even a limited empirical analysis or qualitative discussion of variance could further strengthen the intuition behind the proposed method and its reliability guarantees.
>
> 2. Q5 (Writing and Consistency Issues):
> While several writing issues have been addressed, I noticed that some typographical and consistency problems remain. For example:
>
> a) Hindsight information matching (HIM) should be consistently capitalized as Hindsight Information Matching (HIM), in line with other method names.
>
> b) Expected Calibration Error (ECE)  appears to be defined twice in Section 3.2.
>
> c) Calibration Enhanced Decision Maker (CEDM) is redefined multiple times throughout the text.
>
> I recommend performing another careful pass over the manuscript to eliminate repeated definitions, ensure consistent capitalization, and correct remaining typographical issues. Addressing these points would further improve the readability of the paper.

---

> ### Author Response · Authors · 2025-12-24
>
> Dear Reviewer y6VS:
>
> We would like to thank you once more for your invaluable feedback and precious suggestions on the manuscript. Based on your suggestions, we have made further revisions to the manuscript.
>
> > Q2 (Variance of Calibration Error)
>
> Thank you for your valuable suggestion. We further expand the reporting of standard deviations in the evaluations for different numbers of bins (Appendix E). Specifically, we report the mean evaluation results under different numbers of bins in Tables 16, 17, and 18, with the corresponding standard deviations reported in Tables 19, 20, and 21. Compared with the original paradigm (Tables 1, 2, 3), the CEDM paradigm exhibits lower variance overall.
>
> > Q5 (Writing and Consistency Issues)
>
> Thank you once again for your valuable suggestion. We have thoroughly revised the points. Specifically, we ensure that all method names are defined only once and use consistent capitalization throughout the manuscript. Meanwhile, for EACE and CEDM (two primary concepts of this paper), we retain their abbreviations separately in the Abstract, Introduction, and Methodology sections to avoid potential misunderstanding for readers who may only read the Abstract or Introduction.
>
> Finally, we would like to thank you once again for your invaluable feedback and precious suggestions on the manuscript.
>
> Best regards,
>
> Authors of Submission 6307

---

### Review · Reviewer_KM5j · 2025-11-29

**Summary Of Contributions:**

The Authors provide a procedure for estimating the reliability of sequential  actions for  Offline Deep  Reinforcement Learning. This is important for addressing the reliability of rare occurrences. The Manuscript  show promising results improving the robustness of the trained systems.

The procedure is based on binning the policies according to their frequencies in the training datasets. The experimental histogram of offline data and the trained policy for the same are compared with Wasserstein distance. The distributions of the histograms show how reliable the action sequence is on average, their distance between the distributions show how well the trained data matches the experimental data. To me KL-divergence of the probability histogram would be easier to calculate the distribution distances.

The procedure is mathematically sound and well written.  The results show that it improves the results, but is sensitive length of the frame of actions till RTG.

**Additional Comments:**

None

**Audience:**

Yes

**Audience Explanation:**

The problem of robustness becomes more and more significant and Authors provide a rather simple and well explained method estimate this. However, a topic of interest would be the influence of data set size. It is expected that larger data sets combined with larger neural networks have emergent capabilities.  s this also reflected in the estimators?

**Broader Impact Concerns:**

This Manuscripts provides means to ease ethical problems on rare incident handling of high risk cases.  However, it does not ensure  a guarantee, but indication. The Authors provide an appropriate statement on Broader Impact Concerns

**Claims And Evidence:**

Yes

**Claims Explanation:**

The definition of the procedure is clear, proven and concise providing estimators  Expected Agent Calibration Error and Calibration Enhanced Decision Maker.

**Requested Changes:**

1) The language does not sound English.  Please ask a native speaker to check it out.
Examples:
"remains a paramount concern within the community" -> concerns the community.
"While extant research". -> prior work
"remains a comparatively underexplored avenue of investigation" ->  remains a rather rare topic.
"we pioneer the concept of "  we introduce the concept of
"A series of in-depth case studies are undertaken to examine CEDM ". A series of In-depth case studies of CEDM are carried out, with applications on Decision Transformer, ....

2) I would like to see if using KL-divergence (i.e cross - entropy)  would change the results when used for estimating the similarites of the distributions.

3) What is the effect of the size of data sets on the performance?

---

> ### Author Response · Authors · 2025-12-13
>
> Dear Reviewer KM5j:
>
> We would like to thank you from the bottom of our heart for your time and efforts on our work. And we sincerely thank you for the thoughtful feedback that will surely turn our paper into a better shape. In the new version of our paper, additional parts have been highlighted in ${\color{blue}blue}$. We offer our responses to address the concerns as follows.
>
> >Q1: Language improvement.
>
> Thank you very much for the valuable suggestion. We have carefully revised the manuscript accordingly and have also sought assistance from a native English speaker for further polishing.
>
> >Q2: KL-divergence results.
>
> We add this in Appendix D.2.
>
> >Q3: Evaluations for different size of datasets.
>
> We add this in Appendix F.
>
> Finally, we would like to thank you once again for the constructive suggestions and insightful comments on our work.
>
> Best regards,
>
> Authors of Submission 6307

---

> > ### Comment · Reviewer_KM5j · 2025-12-30
> > **Comments to the new submission.**
> >
> > The Manuscript language has been improved and the manuscript now includes  further studies on the performance of the proposed method in the appendix using KL-divergence and also the impact of the size of the datasets.
> > My concerns are addressed and I can now recommend its acceptance.

---

### Author Response · Authors · 2025-12-13

Dear Action Editor and Reviewers,

We would like to thank you from the bottom of our heart for your time and efforts on our work. In the new version of our paper, additional parts have been highlighted in ${\color{blue} blue}$. In this short note, we summarize the primary additions made to the manuscript.

* In the section of Introduction, we add some part for large sequence models.
* In the section of Methodology, we include a more rigorous theoretical formulation.
* In the Appendix C.1, we add a detailed explanation of the tasks.
* In the Appendix D.1, we present the results of an implementation of EACE.
* In the Appendix D.2, we report the results evaluated using KL divergence.
* In the Appendix D.3, we report the results evaluated using TV divergence.
* In the Appendix E, we report evaluations conducted under different numbers of bins.
* In the Appendix F, we report evaluations conducted under different dataset sizes.
* In the Appendix G, we provide a comparison with Cal-QL and other Q-learning methods.

Finally, we would like to thank you once more for your great efforts and time on our work, for your insightful comments and for your constructive suggestions.

Best regards,

Authors of Submission 6307

---

### Decision · Action_Editor_ANaB · 2025-12-31

**Recommendation:** Accept as is

**Audience:**

Yes

**Audience Explanation:**

Overall, the review consensus supports acceptance: the paper offers a clear conceptual framework, a theoretically motivated metric, a practical and general method, and thorough empirical validation, making it a solid contribution to the offline RL literature.

**Claims And Evidence:**

Yes

**Claims Explanation:**

This paper addresses the problem of agent calibration in offline reinforcement learning, with a particular focus on trajectory-optimization methods based on large sequence models. The reviewers consistently identifyy as underexplored compared to calibration in model-based and model-free offline RL. The authors introduce EACE as a principled metric for evaluating calibration at the agent level, analogous to Expected Calibration Error. To operationalize calibration improvement, the authors propose CEDM, a general architectural augmentation for sequence-model–based offline RL.